

# Finite-size corrections for universal boundary entropy in bond percolation

Jan de Gier[1], Jesper Lykke Jacobsen[2] and Anita Ponsaing[1]

**1** ARC Centre of Excellence for Mathematical and Statistical Frontiers (ACEMS), School of Mathematics and Statistics, The University of Melbourne, VIC 3010, Australia

**2** [a]LPTENS, École Normale Supérieure – PSL Research University, 24 rue Lhomond, F-75231 Paris Cedex 05, France

[b]Sorbonne Universités, UPMC Université Paris 6, CNRS UMR 8549, F-75005 Paris, France

[c]Institut de Physique Théorique, CEA Saclay, F-91191 Gif-sur-Yvette, France

jdgier@unimelb.edu.au, jesper.jacobsen@ens.fr, aponsaing@unimelb.edu.au

## Abstract

We compute the boundary entropy for bond percolation on the square lattice in the presence of a boundary loop weight, and prove explicit and exact expressions on a strip and on a cylinder of size $L$. For the cylinder we provide a rigorous asymptotic analysis which allows for the computation of finite-size corrections to arbitrary order. For the strip we provide exact expressions that have been verified using high-precision numerical analysis. Our rigorous and exact results corroborate an argument based on conformal field theory, in particular concerning universal logarithmic corrections for the case of the strip due to the presence of corners in the geometry. We furthermore observe a crossover at a special value of the boundary loop weight.

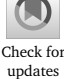

# 1 Introduction

A 2D classical statistical mechanical model can be viewed as a 1+1D model evolving in imaginary time. It is well known that at a critical point the Hamiltonian (or equivalently the transfer matrix) of such a model can be described by conformal field theory (CFT). The Hamiltonian $H_L$ can be related to the CFT translation operator. Consequently, analysing the eigenvalues of $H_L$ on the lattice for different sizes $L$ is a very useful way of extracting the conformal spectrum [1, 2].

It was shown in [3] that scalar products can be measured as well, by mimicking on the lattice the construction of in and out states and scalar products of the continuum limit [4, 5]. Natural scalar products on the lattice were proposed for the examples of the Ising chain and the Temperley–Lieb loop model. Strong numerical evidence in [3, 6, 7] suggests that these lattice scalar products indeed go over to the continuum limit ones as naively expected, for all quantities of interest.

One such quantity of interest is the boundary entropy [8] which is defined in the following way. For a given CFT, one can define several conformally invariant boundary conditions which are encoded by a boundary state [9, 10]. When one perturbs a conformal boundary condition (CBC) by a relevant operator, it flows towards another CBC under the renormalisation group

flow. These CBCs and their flows can be characterised by their boundary entropy $S_B$ which is defined in the CFT via the scalar product of a boundary state $|B\rangle$ with the ground state $|0\rangle$ of the conformal Hamiltonian:

$$S_B = -\log\langle B|0\rangle. \tag{1}$$

These numbers are universal and have been computed analytically for many CFTs and for many different CBCs. In the context of CBCs relevant for loop models [11], several such analytical computations were presented in [12, 13]. Using the lattice regularisation of the scalar products, $S_B$ was also investigated numerically in [3] from finite-size calculations in two examples: the periodic Ising chain and the Temperley–Lieb (TL) loop model on the cylinder.

In this paper we will focus on the TL loop model and provide several mathematically rigorous results on the computation of these scalar products on a lattice of size $L$ and for the case where the bulk loop weight $\beta = 1$. At this value of the loop weight the model is well known to be equivalent to bond percolation [14]. We note here that this model is also equivalent to the stochastic raise and peel model for which the stationary state entanglement entropy in the context of shared information was studied in [15]. The rigorous finite size results allow us to perform a detailed asymptotic analysis in $L$, of which the universal contribution can be compared to the CFT predictions. Moreover, we will compute $S_B$ for the TL loop model on the cylinder as well as on a strip. The latter gives rise to non-trivial universal logarithmic corrections in the CFT due to the existence of corners [6, 7, 16–18].

We note that scalar products, or overlaps, such as (1) have been recently considered in the context of non-equilibrium dynamics in spin chains. The result for the $q$-dimerised boundary state in [19] in the XXZ spin chain is relevant to this paper (the loop weight $\beta$ is related to $q$), and could also be computed using the approach of [20]. Here we follow a very different path in computing the overlap (1), which is more explicit but only valid when $q$ is a third root of unity, or $\beta = 1$.

In the following section we give descriptions of the loop model on a cylinder (for even size $L = 2n$) as well as on a strip, i.e., with periodic and reflecting boundary conditions. We will be precise in terms of mathematical statements, writing "conjecture" for statements we are very confident about being true but for which a rigorous mathematical proof is lacking.

## 1.1 The Temperley–Lieb algebra

The Temperley–Lieb algebra is built from the generators $\{e_i \mid 1 \leq i < L\}$, which satisfy the following relations,

$$e_i^2 = \beta e_i, \qquad e_i e_{i\pm 1} e_i = e_i, \tag{2}$$

with $\beta$ a parameter of the model. This algebra can be supplemented with extra generators that dictate boundary conditions. We will consider two types of boundary conditions, periodic and reflecting (corresponding to the cylinder and the strip). The reflecting case consists simply of the above generators, while the periodic case has an extra generator $e_L$ that satisfies the same relations as the others, working mod $L$:

$$e_L^2 = \beta e_L, \qquad e_L e_1 e_L = e_L, \qquad e_1 e_L e_1 = e_1. \tag{3}$$

In addition to these local relations, in the even periodic case we impose the idempotent relations

$$I_1 I_2 I_1 = I_1, \qquad I_2 I_1 I_2 = I_2, \tag{4}$$
$$I_1 = e_1 e_3 \dots e_{L-1}, \qquad I_2 = e_2 e_4 \dots e_L,$$

to ensure that non-contractible loops going around the cylinder also have weight 1. Similar quotients need to be defined in the odd case [21].

There is a natural representation of this algebra in terms of link patterns. These are perfect matchings of $L$ sites, which satisfy the imposed boundary conditions. If $L$ is odd there will be one site that is unpaired (or connected to a point at infinity). In the periodic case we can view the link pattern either as a chord diagram (see Figure 1a), or as matchings of sites along a strip, similarly to the reflecting case (see Figure 1b), with periodic boundary conditions understood. We use $\text{LP}_L$ to refer to the set of link patterns of size $L$.

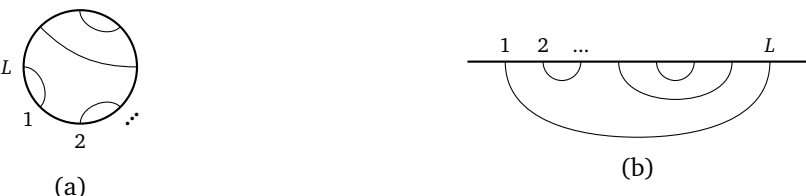

Figure 1: Example periodic and reflecting link patterns for $L = 8$.

In this representation the Temperley–Lieb generator $e_i$ acts between site $i$ and $i + 1$, and has the graphical representation

$$e_i = \quad \underset{\substack{i \quad i+1}}{\overset{\smile}{\frown}} \quad . \tag{5}$$

The algebraic rules (2) amount to the rules-of-thumb "strings are pulled tight, closed loops are replaced with a weight of $\beta$". An example of the action of $e_i$ on a link pattern is

$$e_3 \quad 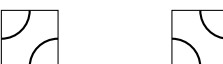 \quad . \tag{6}$$

## 1.2 The Temperley–Lieb loop model

The Temperley–Lieb loop model, or completely packed $O(n)$ loop model, is a model on a square lattice where each face of the lattice has loops drawn on it in one of two possible configurations:

The lattice is arranged either on a semi-infinite cylinder or a semi-infinite strip, depending on the boundary conditions. In the reflecting case, arcs are drawn at the boundaries between neighbouring rows (see Figure 2).

When $\beta = 1$, this lattice model is equivalent to the bond percolation model (or the $Q = 1$ Potts model). With sites located on alternating vertices of the lattice, the loops on a face describe whether or not a bond exists between the two sites on opposite corners. For example:

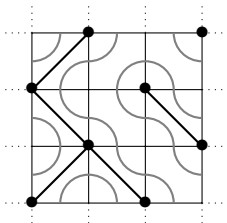

By applying the rules of the Temperley–Lieb algebra, the configurations of loops on the lattice can be grouped according to the link patterns they produce at the top of the lattice. In

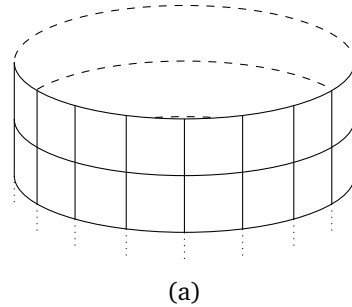 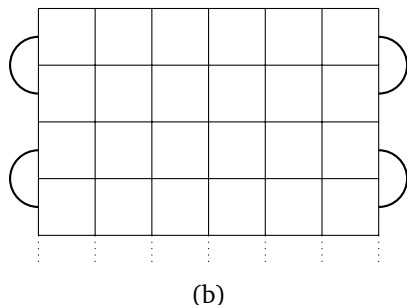

(a)             (b)

Figure 2: Square lattices for the periodic (a) and reflecting (b) Temperley–Lieb loop models. Both cylinder and strip extend downward to infinity.

this way the states of the model live in a vector space with a basis indexed by link patterns, and the model has Hamiltonian

$$H_L^{(\mathrm{per})} = \sum_{i=1}^{L}(1 - e_i), \qquad H_L^{(\mathrm{refl})} = \sum_{i=1}^{L-1}(1 - e_i). \tag{7}$$

For our purposes we consider only the case where $\beta = 1$. In this case the Hamiltonian has a ground state eigenvalue of 0, with trivial left eigenvector

$$\langle \Psi_L | = \sum_{\alpha \in \mathrm{LP}_L} \langle \alpha |, \tag{8}$$

and non-trivial right eigenvector, or ground state,

$$|\Psi_L\rangle = \sum_{\alpha \in \mathrm{LP}_L} \psi_\alpha |\alpha\rangle. \tag{9}$$

The ground state at $\beta = 1$ can be normalised to have integer components, where the smallest component is 1 [22, 23]. The normalisation is simply the sum of components,

$$Z_L = \sum_{\alpha \in \mathrm{LP}_L} \psi_\alpha. \tag{10}$$

We recall that the sum of components $Z_L$ is given by [24, 25]

$$Z_{2n}^{(\mathrm{per})} = \mathrm{A}_n, \qquad Z_{2n}^{(\mathrm{refl})} = \mathrm{AV}_{2n+1}, \qquad Z_{2n+1}^{(\mathrm{refl})} = \mathrm{C}_{2n+2}, \tag{11}$$

where $\mathrm{A}_n$ is the number of $n \times n$ alternating sign matrices (ASMs), $\mathrm{AV}_{2n+1}$ is the number of vertically symmetric ASMs of size $2n + 1$, and $\mathrm{C}_{2n}$ is the number of cyclically symmetric transpose complement plane partitions of size $2n$. These numbers are explicitly given in Appendix A.

## 1.3 The boundary entropy generating function

Let the link pattern $\alpha_0$ consist of small arcs between sites $2i - 1$ and $2i$, and site $L$ unpaired if $L$ is odd. We define the boundary state $\langle B| = \langle \alpha_0 |$, and consider the generating function $F(x)$ defined by placing this boundary state at the top of the lattice, see Figure 3. Any closed loop that passes through the top boundary acquires a weight $x$ and we sum over all possible configurations.

Let $k_\alpha$ denote the number of closed loops produced when the link pattern $\alpha$ is paired with $\alpha_0$. The generating function $F_L(x)$ for $L = 2n$ or $L = 2n + 1$ is then

$$F_L(x) := \frac{\langle \alpha_0 | \Psi_L \rangle_x}{\langle \alpha_0 | \Psi_L \rangle_{x=1}} = \frac{1}{Z_L} \sum_{\alpha \in \mathrm{LP}_L} \psi_\alpha \langle \alpha_0 | \alpha \rangle_x = \frac{1}{Z_L} \sum_{\alpha \in \mathrm{LP}_L} \psi_\alpha x^{k_\alpha} = \sum_{k=1}^{n} \frac{a_{k,n}}{Z_L} x^k, \tag{12}$$

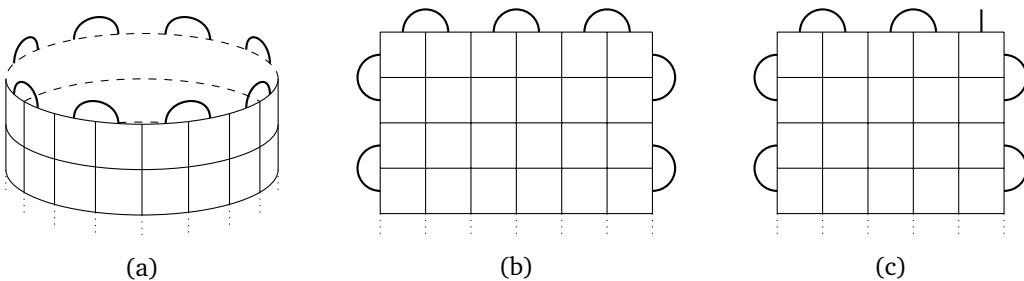

Figure 3: Boundary conditions for $F_L(x)$ in the case of (a) $L = 2n$ periodic, (b) $L = 2n$ reflecting, and (c) $L = 2n + 1$ reflecting.

where $a_{k,n}$ is the sum of components of $\Psi_L$ for which $k_\alpha = k$.

$F(x)$ is the Affleck–Ludwig $g$-factor [8] for critical bond percolation, or rather the Temperley–Lieb (or completely packed O(1)) loop model. For $x = 1$ it is the overlap in the related quantum XXZ spin chain with the deformed dimerised state [19]. One defines the boundary entropy $S_B$ by

$$S_B = -\log\big(F(x)\big). \tag{13}$$

Our aim is to calculate the asymptotics of (13) from (12). This is done mathematically rigorously for the periodic case in Section 4.1, and conjecturally for the reflecting case in Section 4.2.

## 2 Summary of main results

### 2.1 Exact finite size expressions

One of our main results is an explicit expression for any size $L$ for the boundary entropy generating function $F_L(x)$ in both the periodic and reflecting cases of the TL loop model at $\beta = 1$.

**Theorem 1.** *The boundary entropy generating functions $F_L(x)$ on the semi-infinite cylinder and semi-infinite strip are given by*

$$F_{2n}^{(\text{per})}(x) = \sum_{k=1}^{n} \binom{n+k-2}{k-1} \frac{(2n-1)!(2n-k-1)!}{(3n-2)!(n-k)!} x^k, \tag{14}$$

$$F_{2n}^{(\text{refl})}(x) = \prod_{k=0}^{n-1} \left( \frac{(4k+3)!(4k+2)!}{(3k+2)(6k+3)!(2k+1)!} \right) \times$$

$$\det_{1 \le i,j \le n} \left[ \binom{i+j-2}{2j-i} + x \binom{i+j-2}{2j-i-1} \right], \tag{15}$$

$$F_{2n+1}^{(\text{refl})}(x) = \prod_{k=0}^{n} \left( \frac{(4k)!(4k+1)!}{(3k+1)(6k)!(2k)!} \right) \det_{1 \le i,j \le n} \left[ \binom{i+j-1}{2j-i} + x \binom{i+j-1}{2j-i-1} \right]. \tag{16}$$

The proof of this theorem is given in Section 3. Unfortunately we have not been able to obtain an explicit result for odd periodic systems, except for some special values of $x$ which are listed in Appendix A.2.

$F_{2n}^{(\text{per})}(x)$ has the property $F(x) = x^{n+1}F(1/x)$, which is a consequence of the fact that if an even-sized periodic link pattern gives $k$ loops when paired with $\alpha_0$, then its rotation by one step gives $n - k + 1$ loops. Similarly $F_{2n+1}^{(\text{refl})}(x)$ has the property $F(x) = x^n F(1/x)$, which is a consequence of the fact that if an odd-sized reflecting link pattern gives $k$ loops when paired with $\alpha_0$, then its reflection gives $n - k$ loops.

## 2.2 Asymptotics

In order to remove the overall factor of $x$ in the even cases, for both periodic and reflecting boundaries we introduce $\tilde{F}_L(x)$, defined by

$$F_{2n}(x) = x\tilde{F}_{2n}(x), \tag{17}$$

$$F_{2n+1}(x) = \tilde{F}_{2n+1}(x). \tag{18}$$

We have analytically calculated exact asymptotics of $\tilde{F}_L(x)$ as $L \to \infty$ in the periodic case, and conjecture asymptotic expressions in the reflecting case (supported by numerical results and a conformal field theory argument).

### 2.2.1 Asymptotics for the model on the cylinder

To determine the asymptotic behaviour $L \to \infty$ from Theorem 1 it will be convenient to use the parametrisation

$$x = \frac{\sin\left(\frac{\pi(r+1)}{3}\right)}{\sin\left(\frac{\pi r}{3}\right)}, \quad 0 < r < 3. \tag{19}$$

For periodic boundaries and even size $L = 2n$ we can determine the asymptotics of $\tilde{F}_{2n}^{(\text{per})}(x)$ rigorously from (14).

**Proposition 1.** *The asymptotics of $\tilde{F}_{2n}^{(\text{per})}(x)$ is given by*

$$\tilde{F}_{2n}^{(\text{per})}(x) = \varepsilon_{n,x} \exp\left(n f_0(x) + f_1(x) + n^{-1} f_2(x), + \dots\right), \tag{20}$$

*where $\varepsilon_{n,x} = (-1)^{n+1}$ if $x < -1$, $\varepsilon_{n,x} = 1$ otherwise; and*

$$\exp(f_0(x)) = \begin{cases} \dfrac{4}{3\sqrt{3}} \dfrac{1}{\tan\left(\frac{\pi r}{6}\right)} \dfrac{\sin\left(\frac{\pi(r+1)}{6}\right)^2}{\sin\left(\frac{\pi(r+2)}{6}\right)^2}, & 0 < r \le \frac{5}{2} \ (x \ge -1), \\[4mm] \dfrac{-4}{3\sqrt{3}} \dfrac{1}{\tan\left(\frac{\pi(r-3)}{6}\right)} \dfrac{\sin\left(\frac{\pi(r-2)}{6}\right)^2}{\sin\left(\frac{\pi(r-1)}{6}\right)^2}, & \frac{5}{2} \le r < 3 \ (x \le -1), \end{cases} \tag{21}$$

$$\exp(f_1(x)) = \begin{cases} \dfrac{\sqrt{3}}{2} \dfrac{\sin\left(\frac{\pi r}{2}\right)}{\sin\left(\frac{\pi(r+1)}{3}\right)}, & 0 < r \le \frac{5}{2} \ (x \ge -1), \\[4mm] \dfrac{-\sqrt{3}}{2} \dfrac{\sin\left(\frac{\pi(r-3)}{2}\right)}{\sin\left(\frac{\pi(r-2)}{3}\right)}, & \frac{5}{2} \le r < 3 \ (x \le -1), \end{cases} \tag{22}$$

$$f_2(x) = \begin{cases} \dfrac{5}{72}(\cos(\pi r) + 1), & 0 < r \le \frac{5}{2} \ (x \ge -1), \\[4mm] \dfrac{5}{72}(\cos(\pi(r-3)) + 1), & \frac{5}{2} \le r < 3 \ (x \le -1). \end{cases} \tag{23}$$

*At $x = -1$ these expressions are only valid for $n$ odd.*

We note that in each case, the two expressions coincide at $r = 5/2$ ($x = -1$). The first order asymptotics $f_0(x)$ was also calculated in [26, Section 4] ($\tilde{F}_{2n}^{(\text{per})}(x)$ is equal to $h_{2n}(x; \frac{1}{2}, 1)$ in that paper's notation).

### 2.2.2 Asymptotics for the model on the strip

For reflecting boundary conditions we were not able to obtain rigorous results from (15) and (16), except for some special values of the loop weight $x$, see Appendix A.3 and A.4. We can however analyse (15) and (16) with arbitrary numerical precision and have in this way been able to obtain closed form expressions for their exact asymptotics.

We assume the asymptotic form

$$\tilde{F}_L^{(\text{refl})}(x) = \varepsilon_{n,x} \exp\big(n g_0(x) + \log(n) g_1(x) + g_2(x) + n^{-1} g_3(x) + \dots\big), \tag{24}$$

which now contains a logarithmic term that was absent from the periodic case. Here $\varepsilon_{n,x}$ is chosen to match the sign of $\tilde{F}_L^{(\text{refl})}(x)$. This term is due to the presence of corners as will be explained in Section 4.2.

**Conjecture 1.** *With the parametrisation given in* (19)*, for $L = 2n$ we have*

$$g_0 = f_0, \tag{25}$$

$$g_1 = \begin{cases} \frac{1}{6}(1 - r^2), & 0 < r < \frac{5}{2} \ (x > -1), \\ \frac{1}{6}(1 - (r-3)^2), & \frac{5}{2} \leq r < 3 \ (x \leq -1), \end{cases} \tag{26}$$

*and for $L = 2n + 1$,*

$$g_0 = f_0, \tag{27}$$

$$g_1 = \begin{cases} -\frac{1}{6}(1 - r)^2, & 0 < r \leq \frac{5}{2} \ (x \geq -1), \\ -\frac{1}{6}(4 - r)^2, & \frac{5}{2} \leq r < 3 \ (x \leq -1). \end{cases} \tag{28}$$

*(Note that the two expressions for $g_1$ coincide at $x = -1$ in the odd case, but not in the even case, see Figure 4.)*

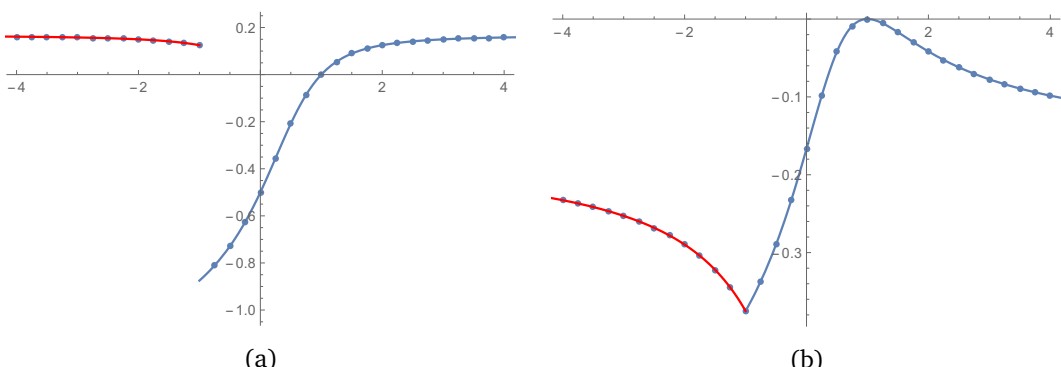

(a)  (b)

Figure 4: Comparison of expressions in Conjecture 1 to numerical results for $g_1(x)$ for (a) $L = 2n$, and (b) $L = 2n+1$, plotted against $x$. In each case, the blue line is the expression for $x > -1$, the red is the expression for $x < -1$, and the blue dots are obtained by a fit to data of $\tilde{F}_L(x)$ from (15) and (16), with even $n$ between 50 and 100.

These formulæ are supported by explicit calculations for the special values of $x$ in Appendix C. While these results are mathematically speaking conjectures, we stress that they can be ascertained with arbitrary numerical accuracy from (15) and (16). They can also be obtained from a conformal field theoretic argument which we will provide in Section 4.2.

# 3 Exact expressions for the boundary entropy for finite size

To prove Theorem 1 we will need a variety of results that are scattered across the literature. In the following we will, where required, clarify the connection between our notation and that used elsewhere. First we elucidate a few different ways to express the number of boundary loops $k_\alpha$ in terms of properties of the link pattern $\alpha$.

## 3.1 Boundary loops and properties of Dyck paths

There is a well-known bijection between link patterns and Dyck paths (see Figures 5 and 6). For even $L$, a Dyck path is a path of $L$ steps from $(0,0)$ to $(L,0)$, where each step can either be a diagonal up-step or a diagonal down-step, such that the height of the path is never less than zero. The bijection to link patterns is made by interpreting each up-step from $(i-1, j-1)$ to $(i, j)$ as an opening of a link at site $i$, and each down-step from $(i-1, j)$ to $(i, j-1)$ as a closing of a link at site $i$. For odd $L$, the interpretation of a link pattern as a Dyck path is slightly different: The unpaired link is interpreted as an up-step and the path ends at height one, instead of height zero (see Figure 6).

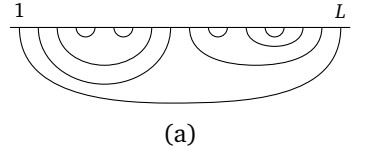
(a)

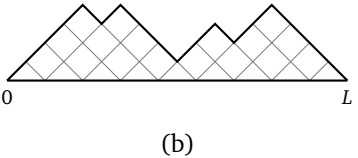
(b)

Figure 5: An even-sized (periodic or reflecting) link pattern (a) and its interpretation as a Dyck path (b).

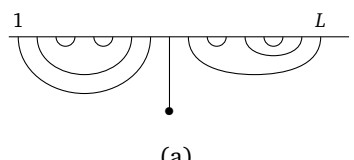
(a)

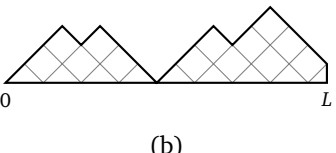
(b)

Figure 6: An odd-sized link pattern (a) and its interpretation as a Dyck path (b).

With respect to Dyck paths, we make two definitions.

**Definition 1.** *Each Dyck path associated with a link pattern $\alpha$ can be filled underneath with tiles as shown in Figure 5b. We define $s_\alpha$ to be the signed sum of these tiles, which is found by assigning $+1$ or $-1$ to a tile depending on its vertical position, starting from $+1$ on the first row of complete (square) tiles, and then summing these weights over all tiles, for example:*

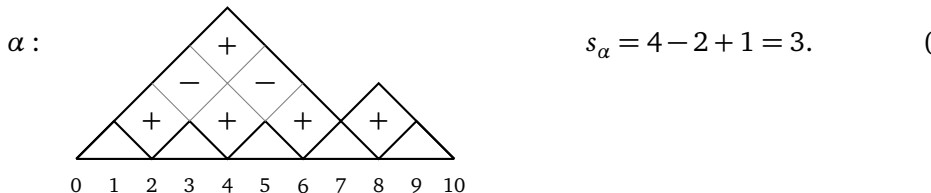

$$s_\alpha = 4 - 2 + 1 = 3. \qquad (29)$$

*Note that the assignment of signs can equally be made in terms of the horizontal position, i.e., $+1$ is assigned to tiles on even sites and $-1$ to tiles on odd sites.*

If we also assign $-1$ to the tiles in the 0th row, we get $s_\alpha - n$ for $L = 2n$, or $s_\alpha - n - 1$ for $L = 2n + 1$.

**Definition 2.** *For a link pattern $\alpha$, we define $d_\alpha$ to be the number of Dyck ribbons in the Dyck path corresponding to $\alpha$ [27]. The Dyck ribbon decomposition works as follows: A maximal ribbon of tiles is shaded inside the Dyck path:*

Step 1:

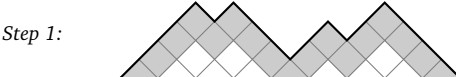

*As can be seen from the picture, it is allowed for a Dyck ribbon to include one of the tiles in the 0th row, but not to cross the horizontal line. After the first Dyck ribbon has been shaded, those tiles are discarded and another ribbon is shaded, and so on, until all the tiles have been discarded.*

Step 2:

Step 3:

Step 4:

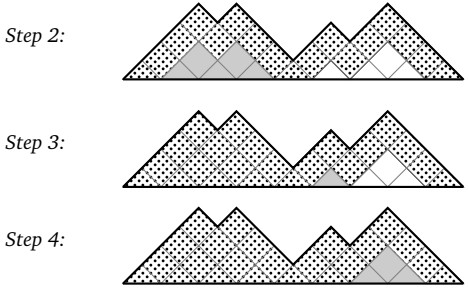

*In this example, $d_\alpha = 4$.*

**Lemma 1.** *For a link pattern of size $L = 2n$ or $L = 2n + 1$, the number of closed loops $k_\alpha$ formed between a link pattern $\alpha$ and $\alpha_0$ is equal to:*

(a) *The number of odd sites in $\alpha$ that are paired to the right,*

(b) *$n - s_\alpha$, and*

(c) *$d_\alpha$ if $L = 2n$; $d_\alpha - 1$ if $L = 2n + 1$.*

*Proof.* (a) Given a link pattern $\alpha$, each site is paired either to the right or the left. Each closed loop formed by the joining of $\alpha$ and $\alpha_0$ passes through exactly one site (denoted $\times$ below) that is paired to the right in both $\alpha$ and $\alpha_0$:

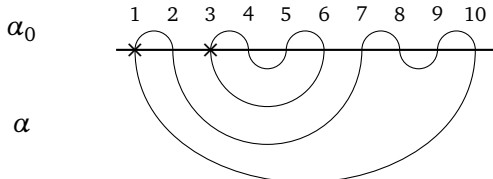

Since the odd sites of $\alpha_0$ are always paired to the right, $k_\alpha$ is just the number of odd sites in $\alpha$ that are paired to the right.

(b) Consider the Dyck path corresponding to $\alpha$. If $L$ is even, each up step in the Dyck path corresponds to a site paired to the right, so $k_\alpha$ is the number of up steps that occur from an even to an odd site (indicated below, remembering that sites in Dyck paths are shifted half a step to the right compared to the link pattern):

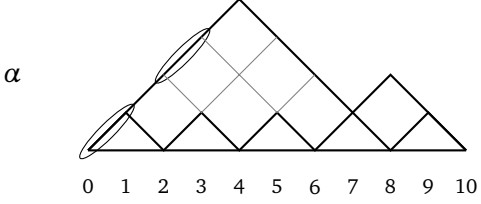

If $L$ is odd, one of the up-steps in the Dyck path corresponds to the unpaired site (always an odd site) in the link pattern, so instead the number of up steps that occur from an even to an odd site is $k_\alpha + 1$.

Recall that the assignment of signs to tiles described in Definition 1, along with the assignment of $-1$ to each half-box in the 0th row, gives a sum of $s_\alpha - n$ for $L = 2n$, $s_\alpha - n - 1$ for $L = 2n + 1$. However we can ignore any 'dominoes' in the alignment

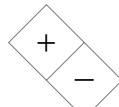

as these contribute 0 to the sum.

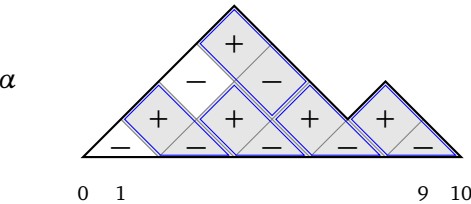

What remains is a collection of tiles of weight $-1$: $n - s_\alpha$ of them for $L = 2n$; $n - s_\alpha + 1$ for $L = 2n + 1$. Each one can be thought of as the bottom right half of a domino, cut off by the Dyck path, which means that each one corresponds to an up step of the path from an even to an odd site. Thus $k_\alpha = n - s_\alpha$.

(c) Consider again the assignment of signs to tiles including the 0th row:

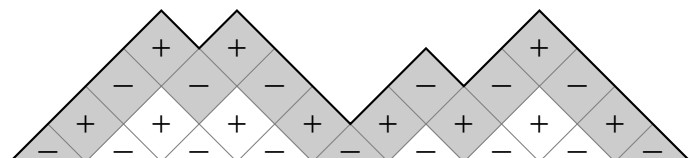

Each Dyck ribbon alternates in sign, beginning and ending on $-1$, thus adding the signs in a Dyck ribbon always gives $-1$. By the end of assignment of Dyck ribbons we have used up all the tiles, so we have

$$-d_\alpha = \begin{cases} s_\alpha - n, & L = 2n, \\ s_\alpha - n - 1, & L = 2n + 1. \end{cases} \tag{30}$$

Thus from part (b), $d_\alpha = k_\alpha$ if $L$ is even, $d_\alpha = k_\alpha + 1$ if $L$ is odd.

$\square$

## 3.2 Periodic boundaries

We will now prove (14). Here $A_n$ is the number of $n \times n$ alternating sign matrices and $A_{n,k}$ is the number of $n \times n$ alternating sign matrices constrained to have a 1 at top of column $k$ (see Appendix A for explicit expressions of these numbers).

**Proposition 2.** *With periodic boundary conditions,*

$$F_{2n}^{(\text{per})}(x) = \sum_{k=1}^{n} \frac{A_{n,k}}{A_n} x^k. \tag{31}$$

*Proof.* In [27, Theorem 2], the deformed staircase Macdonald polynomial

$$M(u_1, \ldots, u_{n-1}; x_1, \ldots, x_n) \tag{32}$$

for the maximally parabolic subgroup of the Iwahori–Hecke algebra is shown to be expressible in terms of the Kazhdan–Lusztig basis, or in other words and for $x_1 = \ldots = x_n = 1$, the components of the ground state of the TL model. For our purposes we set all of the arguments of $M$ to be equal:

$$M(\underbrace{u, \ldots, u}_{n-1}; x_i = 1) = \sum_{\alpha \in \mathrm{LP}_{2n}} c_\alpha \psi_\alpha, \tag{33}$$

with

$$c_\alpha = \left( -\frac{[u]}{[u+1]} \right)^{n-d_\alpha}. \tag{34}$$

From Lemma 1, since $L = 2n$ we know that $d_\alpha = k_\alpha$, so with

$$x = -\frac{[u+1]}{[u]}, \tag{35}$$

we have

$$F_{2n}^{(\mathrm{per})}(x) = \frac{x^n}{Z_{2n}} M(u, \ldots, u). \tag{36}$$

After using (11), we now only need to show that $M(u, \ldots, u) = \sum_{k=1}^{n} A_{n,k} x^{k-n}$. In [27, Section 4.2], it is stated that $M(u, \ldots, u)$ is equal to a constant term expression $A(1, x^{-1}, \ldots, x^{-1})$. It is conjectured in [28, Section 4] and proved in [29] that

$$A\left(1, \frac{1}{x}, \ldots, \frac{1}{x}\right) = N_{10}'\left(\frac{1}{x}, 1, \ldots, 1\right), \tag{37}$$

where $N_{10}'(x^{-1}, t_1, \ldots, t_{n-1})$ is the generating function for a refined counting of totally symmetric self-complementary plane partitions or non-intersecting lattice paths. A constant term formula for $N_{10}'$ is given in [28, eqn (10)]. It is conjectured in the same paper and proved in [30] that

$$N_{10}'\left(\frac{1}{x}, 1, \ldots, 1\right) = \sum_{k=1}^{n} A_{n,k} x^{1-k}. \tag{38}$$

We note that $A_{n,k} = A_{n,n-k+1}$, so we can rewrite this as

$$N_{10}'\left(\frac{1}{x}, 1, \ldots, 1\right) = \sum_{k=1}^{n} A_{n,k} x^{k-n}, \tag{39}$$

completing our proof. □

## 3.3 Reflecting boundaries

Here we prove (15) and (16).

**Proposition 3.** *With $L = 2n$ and reflecting boundary conditions, the generating function $F_{2n}^{(\mathrm{refl})}(x)$ can be written*

$$F_{2n}^{(\mathrm{refl})}(x) = \frac{1}{\mathrm{AV}_{2n+1}} \det_{1 \le i,j \le n} \left[ \binom{i+j-2}{2j-i} + x \binom{i+j-2}{2j-i-1} \right]. \tag{40}$$

*With $L = 2n + 1$ and reflecting boundary conditions, the generating function $F_{2n+1}^{(\mathrm{refl})}(x)$ can be written*

$$F_{2n+1}^{(\mathrm{refl})}(x) = \frac{1}{\mathrm{C}_{2n+2}} \det_{1 \le i,j \le n} \left[ \binom{i+j-1}{2j-i} + x \binom{i+j-1}{2j-i-1} \right]. \tag{41}$$

*Proof.* First consider $L = 2n$. The determinant in (40) appears in [31, eq (6.10)], as $x^n S(2n, n-1|x^{-1})$ with $\beta = \tau = 1$ and $\tilde{p} = 0$ in the notation of that paper. Showing that this is the same determinant is simply a matter of using the binomial identity

$$\sum_s \binom{a}{b-s}\binom{c}{d+s} = \binom{a+c}{b+d}, \tag{42}$$

and performing the sum separately for each term. Our aim is thus to show that $F_{2n}^{(\text{refl})}(x) = x^n S(2n, n-1|x^{-1})/\text{AV}_{2n+1}$.

In that paper $S(t) := S(2n, n-1|t)$ is defined (see [31, eq (6.4)], also [24, eq (5.15)] and [28]) in terms of normalised elements of the homogeneous ground state vector after a basis transformation. (We will avoid details of the full basis transformation here.) The definition of $S(t)$ amounts to

$$S(t) := \sum_{a \in Q_n} t^{m_a} y_a, \tag{43}$$

where $Q_n$ is the set of increasing integer sequences of length $n$, for which $a_1 = 1$ and $a_j \in \{2j-2, 2j-1\}$, $j \in \{2, \dots, n\}$; the exponent $m_a$ is the number of even elements of $a$; and $y_a$ is the element of the transformed ground state corresponding to $a$.

The elements $y_a$ are shown in [31, Lemma 1] to be partial sums of the ground state elements $\psi_\alpha$, which each $\psi_\alpha$ appearing in exactly one $y_a$. The rule that determines which elements $\psi_\alpha$ belong to which partial sum $y_a$ is as follows (see [28, Appendix A]): For each site $i$ of a link opening in $\alpha$, if $i$ is even, then $(\alpha(i)-1) \in a$; if $i$ is odd, then $i \in a$. For example, let $\alpha$ be

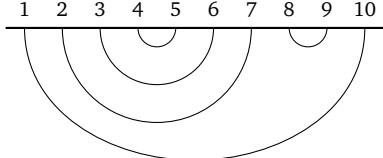

Then $a$ contains $\{1, 6, 3, 4, 8\}$; i.e., $a = (1, 3, 4, 6, 8)$. From this relationship it is also easy to see that the number of odd opening sites is preserved by the basis transformation, so we have $k_\alpha = k_a$ for all $\psi_\alpha$ contributing to $y_a$. In particular we can choose a representative $\alpha$ obtained by interpreting $a \in Q_n$ as a list of starting points of links. The representative set of link patterns obtained from $Q_n$ are those with links nested no more than two deep, or equivalently Dyck paths of length $L$ with height no more than two units. As an example, the sequence $a = (1, 3, 4, 6, 8)$ can be represented by the link pattern or Dyck path shown:

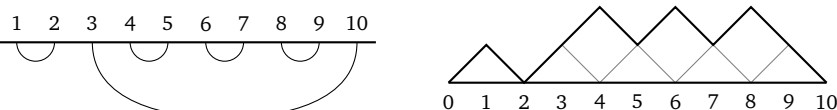

From Lemma 1 we have $m_a = n - k_a$ for these link patterns. Thus, substituting $x = t^{-1}$, we have

$$x^n S(x^{-1}) := \text{AV}_{2n+1} F_{2n}^{(\text{refl})}(x), \tag{44}$$

with the normalisation $Z_{2n} = \text{AV}_{2n+1}$ from (11).

Now consider $L = 2n+1$. The determinant in (41), like the one for the even case, appears in [31, eq (6.19)] as $x^n S(2n+1, n|x^{-1})$, with $\beta = \tau = 1$ and $p = n$, where $S$ is defined in [31, eq (6.15)].[1] The proof given for the even case carries through to the odd case with very few changes (care must be taken with the notation for the new basis elements $a$).  $\square$

---

[1]Note that the determinant form of $N_8(2n; \beta)$ appearing in [32, eq (4.2)] is the specialisation $S(2n-1, n-1|1)$ with general $\beta$. This implies that $N_8(2n; \pm 1) = F(\pm 1)$.

## 4 Asymptotics

### 4.1 Exact asymptotics for periodic boundaries

Recalling (17), first we observe that $\tilde{F}(x) = \tilde{F}_{2n}^{(\text{per})}(x)$ is given in terms of a truncating hypergeometric series,

$$\tilde{F}(x) = \frac{(2n-1)!(2n-2)!}{(n-1)!(3n-2)!} \, {}_2F_1(1-n, n; 2-2n, x), \tag{45}$$

and hence satisfies the following hypergeometric differential equation (this was also observed in [33]):

$$x(x-1)\tilde{F}''(x) + 2(n-1+x)\tilde{F}'(x) - n(n-1)\tilde{F}(x) = 0, \tag{46}$$

with the following initial conditions (given in Appendix A.1):

$$\tilde{F}(1) = 1, \qquad\qquad \tilde{F}(0) = \frac{A_{n-1}}{A_n},$$

$$\tilde{F}(-1) = \begin{cases} 0, & n \text{ even}, \\ \dfrac{AV_n^2}{A_n}, & n \text{ odd}, \end{cases} \qquad\qquad \lim_{x \to \pm\infty} \frac{\tilde{F}(x)}{x^{n-1}} = \frac{A_{n-1}}{A_n}. \tag{47}$$

The asymptotics of the initial conditions can be obtained from the explict form of $A_n$ and $AV_n$ in Appendix A and the asymptotics of Barnes' $G$-function:

$$\log \frac{A_{n-1}}{A_n} = \log\left(\frac{16}{27}\right)n + \log\left(\frac{3\sqrt{3}}{4}\right) + \frac{5}{36n} + \mathcal{O}(n^{-2}),$$

$$\log \frac{AV_n^2}{A_n} = \log\left(\frac{2}{3\sqrt{3}}\right)n + \log\sqrt{6} + \frac{5}{72n} + \mathcal{O}(n^{-3}). \tag{48}$$

Since we are interested in the asymptotics of $\log\tilde{F}(x)$, we assume an expansion of the form

$$\tilde{F}(x) = \exp\left(\sum_{j\geq 0} n^{1-j} f_j(x)\right). \tag{49}$$

Note that the expansion (49) assumes that $\tilde{F}(x)$ is positive so that the $f_j$ are real. This is obviously true for $x \geq 0$, but when $x \to -\infty$ it is easy to see from (14) that the function $\tilde{F}_{2n}^{\text{per}}(x)$ is positive for odd $n$ and negative for even $n$ (Figure 7 shows typical graphs of $\tilde{F}_{2n}(x)$ for even and odd $n$, which demonstrate this). We will deal with this below.

Substituting (49) in (46) and expanding using the small parameter $n^{-1}$, we derive differential equations for the functions $f_j$ with initial conditions given by the coefficients in $n$ of (47)–(48). The first few DEs are:

$$\begin{aligned}
0 &= x(1-x)(f_0')^2 - 2f_0' + 1, \\
0 &= 2f_1'\big(x(1-x)f_0' - 1\big) + (1-x)(2f_0' + xf_0'') - 1, \\
0 &= 2f_2'\big(x(1-x)f_0' - 1\big) + (1-x)\big(2f_1' + xf_1'^2 + xf_1''\big), \\
&\quad \text{etc.} \tag{50}
\end{aligned}$$

The solution to each DE relies on the solution to the previous ones, but for $j \geq 1$ they are simply linear first order DEs. We give here the results for $j = 0, 1, 2$ — the process can be continued to calculate arbitrarily many terms.

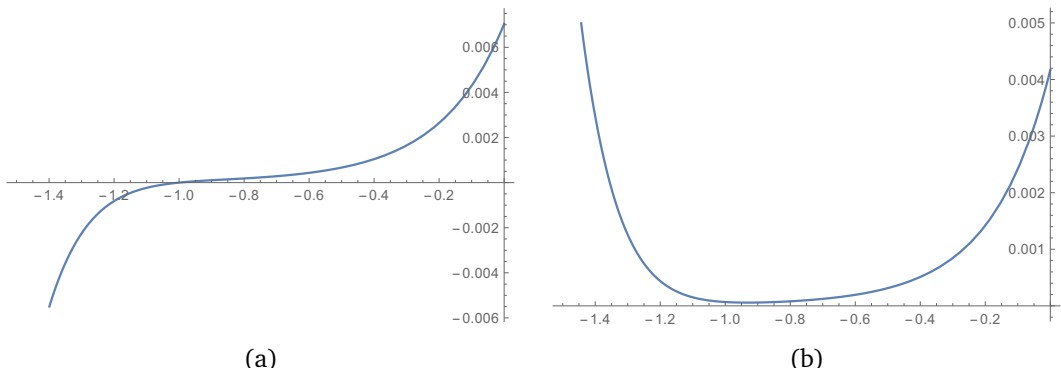

Figure 7: Plot of $\tilde{F}_{2n}(x)$ with (a) $n = 10$, and (b) $n = 11$.

For $j = 0$ the DE has two branches

$$f_0'(x) = \frac{1 \pm \sqrt{1 - x + x^2}}{x(1 - x)}, \tag{51}$$

and the special values

$$f_0(1) = 0, \qquad f_0(0) = \lim_{x \to \pm\infty} (f_0(x) - \log|x|) = \log\left(\frac{16}{27}\right). \tag{52}$$

Only the negative root of (51) is compatible with the boundary condition at $x \to \infty$, as well as with the special values at $x = 0$ and $x = 1$, but it is not compatible with $x \to -\infty$ where the positive root of (51) must be chosen. We further note that since the positive branch of (51) has a pole at $x = 0$, the solution for $x < 0$ that matches the asymptotic boundary condition at $-\infty$ is only valid on $(-\infty, 0)$.

The positive root in (51) satisfies $f_0'(x) < 0$ for $x < 0$, so this branch of $f_0(x)$ is monotone and decreasing. The negative root satisfies $f_0'(x) > 0$ for all $x \in \mathbb{R}$ and hence this branch is monotone and increasing. Furthermore, the range of (the real part of) $f_0(x)$ is $\mathbb{R}$ and therefore there is a special point $x = x_c < 0$ where the branches meet and where $f_0(x)$ is not differentiable. We will prove below that $x_c = -1$.

This can be seen in Figure 8, where data from numerical analysis of $\tilde{F}(x)$ (for odd $n$) are compared with the expressions from Proposition 1. Note the strange location of the data point for $f_1(-1)$. This is related to the non-differentiability of $f_0(x)$ at $x = -1$, and to the separate values for $\tilde{F}(-1)$ for odd and even $n$, see (47). There is a cusp at $x = -1$ in the graph of $\log(\tilde{F}(x)) - nf_0(x)$ (odd $n$), whose width tends to zero as $n \to \infty$, leaving the point at $x = -1$ isolated.

The rest of this analysis will consider $x > x_c$ and $x < x_c$ separately.

### 4.1.1 $x > x_c$

To find $f_0$ it is convenient to parametrise $x$ by (19):

$$x = \frac{\sin(\frac{\pi(r+1)}{3})}{\sin(\frac{\pi r}{3})}, \qquad 0 < r < 3, \tag{53}$$

and we have

$$\sqrt{1 - x + x^2} = \frac{\sqrt{3}}{2\sin(\frac{\pi r}{3})}, \qquad 1 - x = \frac{\sin(\frac{\pi(r-1)}{3})}{\sin(\frac{\pi r}{3})}. \tag{54}$$

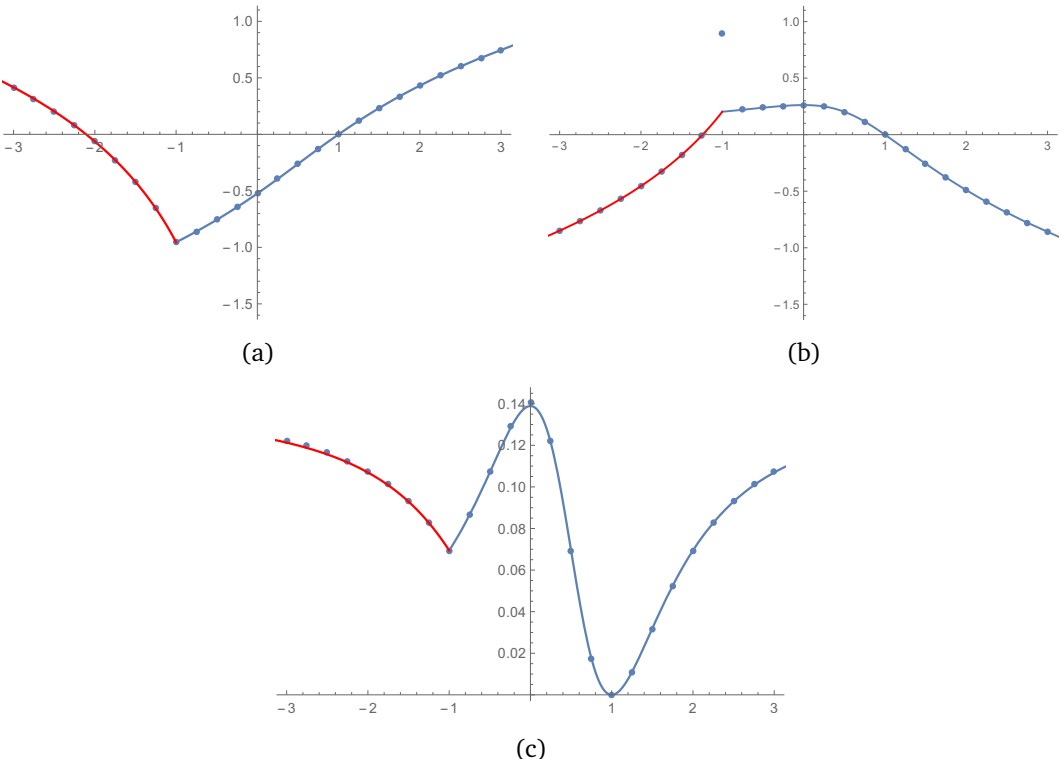

Figure 8: Comparison of expressions in Proposition 1 to numerical results for (a) $f_0(x)$, (b) $f_1(x)$, and (c) $f_2(x)$, plotted against $x$. In each case, the blue line is the expression for $x > -1$, the red is the expression for $x < -1$, and the blue dots are obtained by a fit to data of $\tilde{F}_{2n}(x)$ from (14), with odd $n$ between 101 and 200.

Taking the negative root of (51), which is compatible with $f_0(1) = 0$, we get

$$\frac{d}{dx} f_0(x(r)) = \frac{2\sin\left(\frac{\pi r}{3}\right)}{2\sin\left(\frac{\pi r}{3}\right) + \sqrt{3}}. \tag{55}$$

The extensive boundary entropy may be obtained by integrating this expression, to get

$$\exp\left(f_0(x)\right) = \frac{4}{3\sqrt{3}} \frac{1}{\tan\left(\frac{\pi r}{6}\right)} \frac{\sin\left(\frac{\pi(r+1)}{6}\right)^2}{\sin\left(\frac{\pi(r+2)}{6}\right)^2}. \tag{56}$$

From (50) the equation for $f_1$ reads

$$f_1'(x) = \frac{(1-x)(2f_0' + xf_0'') - 1}{2\left(1 - x(1-x)f_0'\right)}. \tag{57}$$

With the above result for $f_0$, this can be integrated with the initial condition $f_1(1) = 0$, and we obtain

$$\exp\left(f_1(x)\right) = \frac{\sqrt{3}}{2} \frac{\sin\left(\frac{\pi r}{2}\right)}{\sin\left(\frac{\pi(r+1)}{3}\right)}. \tag{58}$$

Likewise, from (50) the equation for $f_2$ reads

$$f_2'(x) = \frac{(1-x)(2f_1' + xf_1'^2 + xf_1'')}{2\left(1 - x(1-x)f_0'\right)}, \tag{59}$$

and with the results for $f_0$ and $f_1$, this can be integrated with the initial condition $f_2(1) = 0$ to give

$$f_2(x) = \frac{5}{72}(\cos(\pi r) + 1). \tag{60}$$

### 4.1.2 $x < x_c$

As mentioned above, $\tilde{F}_{2n}^{\text{per}}(x)$ for even $n$ is negative when $x < x_c$. So for this case we take the expansion

$$\tilde{F}(x) = -\exp\left(\sum_{j\geq 0} n^{1-j} f_j(x)\right), \tag{61}$$

whereas for $n$ odd we take the expansion (49) as before. The boundary condition $x \to -\infty$ gives:

$$\frac{A_{n-1}}{A_n} = \lim_{x\to-\infty} \frac{\tilde{F}(x)}{x^{n-1}} = \lim_{x\to\infty} \frac{\tilde{F}(-x)}{(-x)^{n-1}} = \begin{cases} \lim\limits_{x\to\infty} \dfrac{-\tilde{F}(-x)}{x^{n-1}}, & n \text{ even}, \\[2mm] \lim\limits_{x\to\infty} \dfrac{\tilde{F}(-x)}{x^{n-1}}, & n \text{ odd}. \end{cases} \tag{62}$$

Clearly the resulting boundary conditions for the $f_j$ will therefore be the same in both cases:

$$\lim_{x\to\infty} \left(f_0(-x) - \log(x)\right) = \log\left(\frac{16}{27}\right), \tag{63}$$

$$\lim_{x\to\infty} \left(f_1(-x) + \log(x)\right) = \log\left(\frac{3\sqrt{3}}{4}\right), \tag{64}$$

$$\lim_{x\to\infty} f_2(-x) = \frac{5}{36}. \tag{65}$$

To find $f_0$ we use the same parametrisation as before (53), and take now the positive root of (51). We thus have

$$\frac{d}{dx} f_0(x(r)) = \frac{2\sin\left(\frac{\pi r}{3}\right)}{2\sin\left(\frac{\pi r}{3}\right) - \sqrt{3}}, \tag{66}$$

and integrating this we get

$$\exp\left(f_0(x)\right) = \frac{-4}{3\sqrt{3}} \frac{1}{\tan\left(\frac{\pi(r-3)}{6}\right)} \frac{\sin\left(\frac{\pi(r-2)}{6}\right)^2}{\sin\left(\frac{\pi(r-1)}{6}\right)^2}. \tag{67}$$

The DE for $f_1$ is the same as for $x > x_c$ (57). Using the new result for $f_0$ and integrating we get

$$\exp\left(f_1(x)\right) = \frac{-\sqrt{3}}{2} \frac{\sin\left(\frac{\pi(r-3)}{2}\right)}{\sin\left(\frac{\pi(r-2)}{3}\right)}. \tag{68}$$

Similarly the DE for $f_2$ is the same as for $x > 1$ (59). Using the new results for $f_0$ and $f_1$ and integrating we get

$$f_2(x) = \frac{5}{72}(\cos(\pi(r-3)) + 1). \tag{69}$$

Finally, the value of $x_c$ is obtained by equating (56) and (67), which results in $r_c \in \frac{1}{2} + \mathbb{Z}$. As $x_c < 0$ there is only one solution, namely $r_c = \frac{5}{2}$, for which $x_c = -1$.

## 4.2 Conformal field theory argument for reflecting boundaries

The term $ng_0(x)$ in the exponential expansion (24) of $\tilde{F}_L(x)$ can be interpreted as the surface free energy associated to the particular boundary condition imposed on the top of the strip (see Figure 3). This term is not affected by changing from periodic to reflecting boundary conditions, so $g_0 = f_0$. For the usual reflecting boundary conditions (i.e., $x = 1$ and $r = 1$) we find $g_0 = 0$, as we should, since the generating function is trivial in that case: $\tilde{F}_L(1) = 1$ (implying $g_i(1) = 0$ for all $i \geq 0$).

The next term, $\log(n)g_1(x)$, is universal and the coefficient $g_1(x)$ can be derived using arguments of conformal field theory. To be more precise, CFT will provide an expression for $g_1(x(r))$ that is valid for $r$ within a domain $\mathcal{D}_0$ that contains the trivial point $r = 1$. By general arguments, the same analytical expression $g_1(r)$ should hold at least for positive values of $x$, so $\mathcal{D}_0 \supseteq (0, 2)$. However, since we have parameterised $x$ by (53), which is insensitive to shifting $r$ by multiples of 3, it is possible that the whole interval $r \in (0, 3)$ will be divided into several domains, among which the expression for $g_1(x)$ varies by such shifts in $r$. This is precisely what we see in Conjecture 1, according to which $\mathcal{D}_0 = (0, \frac{5}{2})$. There then exists another domain, $\mathcal{D}_1 = (\frac{5}{2}, 3)$ covering the remainder of the interval $(0, 3)$, on which the analytical expression for $g_1(r)$ is obtained from that on $\mathcal{D}_0$ through the shift $r \to r - 3$.

We now give CFT arguments in support of Conjecture 1 on the domain $\mathcal{D}_0$. Consider first a CFT defined in the upper half plane with an operator of conformal weight $h$ inserted at the origin. This produces a singularity in the stress-energy tensor close to the origin:

$$T(w) \approx \frac{h}{w^2}. \tag{70}$$

We can produce a $\frac{\pi}{2}$ corner at the origin by applying the conformal mapping $z = w^{1/2}$. Recall the usual transformation law

$$T(z) = T(w)\left(\frac{dw}{dz}\right)^2 + \frac{c}{12}\{w; z\}, \tag{71}$$

where $c$ is the central charge of the CFT, and $\{w; z\}$ denotes the Schwarzian derivative. In the new geometry we therefore have the singularity

$$T(z) \approx \frac{2\tilde{h}}{z^2}, \tag{72}$$

where

$$\tilde{h} := 2h - \frac{c}{16} \tag{73}$$

denotes the effective conformal weight at the corner.

The anomaly (72) implies a non-trivial scaling dependence of physical quantities [16, 18], which manifests itself even in the case $h = 0$ when there is no boundary condition changing (BCC) operator residing in the corner (provided that $c \neq 0$). We begin by focussing on this case. In particular, consider the deformed free bosonic theory (Coulomb gas), which describes the continuum limit of the Temperley–Lieb loop model [34]. Parameterising the loop weight as

$$\beta = 2\cos\left(\frac{\pi}{p+1}\right), \tag{74}$$

with $p \in (1, \infty)$, the corresponding central charge is

$$c = 1 - \frac{6}{p(p+1)}. \tag{75}$$

One may compute the continuum limit partition function $Z_{\mathscr{R}}$ of this CFT on a large $L \times M$ rectangle [35,36]. The result is [6, Section III.D]

$$Z_{\mathscr{R}}(L,M) = L^{-4\tilde{h}} Z_{\text{CFT}}(\tau), \tag{76}$$

where the second factor

$$Z_{\text{CFT}}(\tau) = \eta(\tau)^{-c/2} \tag{77}$$

is expressed in terms of the Dedekind function $\eta(\tau)$ and the modular parameter (aspect ratio) $\tau = iM/L$. The first factor in (76) meanwhile picks up an anomaly $L^{-\tilde{h}}$ from each of the four corners.

The result (76) has been confirmed by a large number of explicit computations, including various scalar products and careful derivations for free bosonic and fermionic systems [6, Section IV]. More importantly in the present context, the expression for the corner anomaly has been shown to hold also in the general case, where each corner supports various types of BCC operators [7]. It follows that the universal amplitude takes the general form

$$g_1(x) = -\sum_i \tilde{h}_i, \tag{78}$$

where the sum is over the effective conformal weights (73) of each $\frac{\pi}{2}$ corner.

It thus remains to identify the nature of the BCC operators in the two corners along the top rim of the semi-infinite strip (see Figure 2b). Consider first the even case, $L = 2n$. The reflecting boundary conditions along the left and right sides of the strip amount to giving a weight to loops touching those sides equal to the bulk loop weight $\beta$. This corresponds to free boundary conditions in the equivalent $Q = \beta^2$ state Potts model. The different weight $x$ given to loops touching the top rim of the strip corresponds to the insertion of a BCC operator $\phi_{r,r}$ in the upper-left corner, and another, identical, BCC operator in the upper-right corner that changes back to free boundary conditions along the right side of the strip. With the parametrisation (53), the conformal weight of either operator is found [11] to be $h = h_{r,r}$, where we have used the Kac table notation

$$h_{r,s} = \frac{(r(p+1) - sp)^2 - 1}{4p(p+1)}, \tag{79}$$

and $p$ has the same meaning as in (74). Specialising now to percolation (i.e., $\beta = 1$ and $p = 2$, whence $c = 0$), we obtain from (73) and (78)

$$g_1 = -2(h_{r,r} + h_{r,r}) = \frac{1 - r^2}{6}, \tag{80}$$

in agreement with Conjecture 1.

In the odd case, $L = 2n + 1$, the BCC operator in the upper-left corner is the same, namely $\phi_{r,r}$, but in the upper-right corner there is an additional operator that absorbs the unpaired loop strand (see Figure 6a). This is well known [37] to correspond to the operator $\phi_{1,2}$ in Kac notation. This has to be fused with the other $\phi_{r,r}$ operator. A priori there are two fusion channels,

$$\phi_{r,r} \times \phi_{1,2} = \phi_{r,r-1} + \phi_{r,r+1}, \tag{81}$$

but to obtain the correct result in the limit $r \to 1$, when $\phi_{r,r} = \phi_{1,1}$ is the identity operator, the only tenable option is $\phi_{r,r+1}$. Specialising again to percolation, we thus have

$$g_1 = -2(h_{r,r} + h_{r,r+1}) = \frac{-(1-r)^2}{6}, \tag{82}$$

which again agrees with Conjecture 1.

### 4.3   Conformal field theory argument for the even periodic case

The term $f_1(x)$ appearing in the asymptotic expansion of $\tilde{F}_{2n}^{(\mathrm{per})}(x)$ can be rederived by CFT arguments as well. The universal part of the boundary entropy $S_B$ in (13) is the coefficient of the constant, $n$-independent term. We therefore have to be careful with normalisations, in particular regarding the extra factor of $x$ appearing in (17). Let us write

$$S_B^{(\mathrm{univ})} = -\log g_{\mathrm{AL}} \tag{83}$$

for the universal part of $S_B$, where $g_{\mathrm{AL}}$ is the so-called Affleck-Ludwig $g$-factor [8]. As in the preceeding section we take the boundary loop weight $x$ parametrised in terms of $r$ as in (53). The CFT argument will then hold for $r$ inside the domain $\mathscr{D}_0 = (0, \frac{5}{2})$, where we have from Proposition 1

$$g_{\mathrm{AL}} = x \exp(f_1(x)) = \frac{\sqrt{3}}{2} \frac{\sin\left(\frac{\pi r}{2}\right)}{\sin\left(\frac{\pi r}{3}\right)} . \tag{84}$$

We now outline the CFT derivation of this result, following [3, 13, 38].

Consider the continuum limit of our model defined on a cylinder of *finite* height $m$ and circumference $n$, cf. Figure 3a. The top of the cylinder is endowed with the boundary conditions $|b\rangle$ defining the special loop weight $x$, whereas the bottom sustains the usual reflecting boundary conditions $|a\rangle$ with $x = \beta$. The following argument applies for any value $\beta$ of the bulk loop weight inside the critical range, $\beta \in [0, 2]$.

According to the principle of modular invariance, there are two equivalent ways of writing the corresponding continuum-limit partition function $Z_{ab}(m, n)$, corresponding to two different quantisation schemes. In the first scheme, we build the cylinder using a time-evolution operator $U^{(\mathrm{per})} = \mathrm{e}^{-H^{(\mathrm{per})}}$ that propagates the system upwards from the initial state $|a\rangle$ to the final state $\langle b|$. This reads

$$Z_{ab}(m, n) = \left\langle b \middle| (U^{(\mathrm{per})})^m \middle| a \right\rangle = \left\langle b \middle| \tilde{q}^{L_0 + \bar{L}_0 - \frac{c}{12}} \middle| a \right\rangle, \tag{85}$$

where we have introduced the (conjugate) modular parameter $\tilde{q} = \mathrm{e}^{-2\pi m/n}$, the Virasoro generators $L_0$ and $\bar{L}_0$, and the central charge $c$. The Hamiltonian for the periodic system (closed string channel) then reads $H^{(\mathrm{per})} = \frac{2\pi}{n}(L_0 + \bar{L}_0 - \frac{c}{12})$. In the second scheme, the time-evolution operator $U^{(\mathrm{open})}$ propagates the system horizontally between the boundary conditions $a$ and $b$.[2] The partition function is then a trace (and more precisely a Markov trace, due to the non-local nature of the loop weights):

$$Z_{ab}(m, n) = \mathrm{Tr}_{ab}\left(U^{(\mathrm{open})}\right)^n = \mathrm{Tr}_{ab}\left(q^{L_0 - \frac{c}{24}}\right), \tag{86}$$

where now $U^{(\mathrm{open})} = \mathrm{e}^{-H^{(\mathrm{open})}}$, and the Hamiltonian for the non-periodic system (open string channel) reads $H^{(\mathrm{open})} = \frac{\pi}{m}(L_0 - \frac{c}{24})$. Note that this involves only a chiral CFT; the corresponding modular parameter is $q = \mathrm{e}^{-\pi n/m}$.

The expression for $Z_{ab}(m, n)$ in the second scheme has been established in [11], in a more general situation where the weights of loops depend on their homotopy class. Let $\beta = 2\cos\gamma$ (resp. $x$) be the weight of loops homotopic to a point that do not touch (resp. touch) the $b$-boundary. Similarly, let $\ell = 2\cos\chi$ (resp. $\ell_1 = \frac{\sin(u+1)\chi}{\sin u\chi}$) be the weight of non-contractible loops (i.e., loops that wind around the cylinder) and that do not touch (resp. touch) the $b$-boundary. Here we use convenient parametrisations in terms of parameters $\chi$ and $u$. Finally, let $g = 1 - \frac{\gamma}{\pi}$ denote the Coulomb gas coupling constant [34]. The result of [11] then reads

$$Z_{ab}(m, n) = \frac{q^{-c/24}}{P(q)} \sum_{j \in \mathbb{Z}} \frac{\sin(u + 2j)\chi}{\sin u\chi} q^{h_{r,r+2j}} , \tag{87}$$

---

[2]On the lattice these are implemented using boundary Temperley-Lieb algebras [39, 40]

where $P(q) = \prod_{k=1}^{\infty}(1-q^k)$, and $h_{r,s}$ refer to the conformal weights (79), here with $\gamma = \frac{\pi}{p+1}$. The label $j$ corresponds to the sector with $|2j|$ non-contractible loops, of which the uppermost touches (resp. does not touch) the $b$-boundary for $j > 0$ (resp. for $j < 0$). The corresponding amplitude can be written [38, eq. (41)]

$$\frac{\sin(u+2j)\chi}{\sin u\chi} = \ell_1 U_{2j-1}\left(\frac{\ell}{2}\right) - U_{2j-2}\left(\frac{\ell}{2}\right) \tag{88}$$

where $U_k(z)$ is the $k$th order Chebyshev polynomial of the second type. The amplitude was first found in the latter form by using a rigorous combinatorial approach [41], in which the Markov trace was decomposed on usual (matrix) traces within each standard module corresponding to the label $j$.

Using the Poisson summation formula, the expression (87) can now be transformed into the first quantisation scheme, that is, in terms of the parameter $\tilde{q}$. The result is [38, eq. (42)]

$$Z_{ab}(m,n) = (2g)^{-1/2}\frac{\tilde{q}^{-c/12}}{P(\tilde{q}^2)}\sum_{p\in\mathbb{Z}}\frac{\sin\left(u\chi + r\frac{\gamma}{g}(p+\frac{\chi}{\pi})\right)}{\sin u\chi}\tilde{q}^{\frac{1}{2g}\left[\left(\frac{\chi}{\pi}+p\right)^2-\left(\frac{\gamma}{\pi}\right)^2\right]}. \tag{89}$$

In this form we can now take the limit $m \to \infty$ of a half-infinite cylinder. In that limit $\tilde{q} \ll 1$, and the dominant contribution to (89) comes from the $p = 0$ term (where the eigenvalue of $L_0 + \bar{L}_0$ is the trivial critical exponent $h_0 + \bar{h}_0 = 0$). We have then

$$Z_{ab}(m,n) \sim \langle b|0\rangle\langle 0|a\rangle\, e^{\frac{\pi c}{6}\frac{m}{n}}, \tag{90}$$

where $|0\rangle$ denotes the ground state of the CFT. Finally, we identify the scalar product of this with the boundary state as $g_{\mathrm{AL}} = \langle b|0\rangle$, whereas $\langle 0|a\rangle$ is the same quantity evaluated at $r = 1$ (reflecting boundary conditions). Thus, setting $\chi = \gamma$ and $u = r$ for simplicity, we obtain

$$\langle b|0\rangle\langle 0|a\rangle = (2g)^{-1/2}\frac{\sin\frac{r\gamma}{g}}{\sin r\gamma}, \tag{91}$$

and from this one deduces [3, eq. (13)]

$$g_{\mathrm{AL}} = (2g)^{-1/4}\frac{\sin\frac{r\gamma}{g}}{\sin r\gamma}\left(\frac{\sin\gamma}{\sin\frac{\gamma}{g}}\right)^{1/2}. \tag{92}$$

Specialising now to the case of bond percolation ($\gamma = \frac{\pi}{3}$ and $g = \frac{2}{3}$) this reproduces (84) indeed.

The result (92) was checked against numerical evaluations of the lattice scalar product in [3, figure 2] for several values of $\beta$, including $\beta = 1$, finding in all cases excellent agreement. It should be stressed that in [3] the square lattice was turned by an angle $\frac{\pi}{4}$ with respect to our conventions, so the agreement found demonstrates that $g_{\mathrm{AL}}$ is indeed universal, i.e., independent of details of the lattice realisation.

## 5  Conclusion

We have computed the overlap of the ground state of the Temperley-Lieb loop model with bulk loop weight $\beta = 1$ with that of the product state of small arcs, or deformed dimerised state. We have done so on the cylinder as well as on the strip, and computed the generating function $F(x)$ in those cases by giving a weight $x$ to boundary loops.

The boundary entropy $S_B$ for critical bond percolation can be found from the generating function $F(x)$ via the formula

$$S_B = -\log\big(F(x)\big). \tag{93}$$

We have calculated $F(x)$ rigorously for finite sizes in the even-sized periodic case as well as the even- and odd-sized reflecting cases. In the periodic case we have derived exact asymptotics as a function of $x$, which agrees with the predictions of CFT, and in the reflecting case we have made a conjecture for the subleading $\log(n)$ term in the exponent based on CFT arguments and supported by numerical data of arbitrary high precision.

Clearly a finite-size expression for the odd-sized periodic case is still lacking. We have collected some small-size data to aid the search in Appendix B. However our results for the leading-order asymptotics of the periodic system should not depend on the parity of the system size.

We would like to be able to rigorously calculate exact asymptotics for the reflecting case as well, but the form of the finite-size expression (a determinant of a matrix whose size grows with the lattice size) presents difficulties. The usual techniques to compute asymptotics from determinants do not seem obviously applicable, and further exploration is out of the scope of this paper. We have been able, however, to obtain and conjecture exact analytic expressions in this case based on high-precision numerical analysis of the finite-size expressions.

Finally, the quantity $F(x)$ appears to hold many combinatorial secrets. This can be gathered from the various conjectures in Appendix A.3 and the small-size examples in Appendix B. The most intriguing of these combinatorial connections is the following: The Razumov–Stroganov–Cantini–Sportiello Theorem [42, 43] gives an interpretation of the even periodic TL ground state components $\psi_\alpha$ in terms of alternating sign matrices. This interpretation implies that the numbers $A_{n,k}$ from (31) are not only the numbers of ASMs refined according to the position of the 1 in the top row, but also a *different* refined counting of ASMs (the number of closed loops through the top boundary, $k$, carries through to an equivalent statistic on the ASM side). Thus there is an equivalence between two separate refined countings of ASMs — a purely combinatorial result, proved via the TL loop model. It would be interesting to find a combinatorial proof of this result.

# Acknowledgments

We warmly thank Filiberto Ares, Roger Behrend, Filippo Colomo, Jérôme Dubail, György Fehér, Anthony Mays, Bernard Nienhuis, Amilcar Rabelo de Queiroz, Hjalmar Rosengren, Michael Wheeler and Paul Zinn-Justin for stimulating discussions. JdG and AP thank the Australian Research Council for generous financial support. JJ is supported by the Institut Universitaire de France, and by the European Research Council through the Advanced Grant NuQFT.

# A  Combinatorial numbers

We use the following product formulæ:

$$
\begin{aligned}
A_n &= \prod_{j=0}^{n-1} \frac{(3j+1)!}{(n+j)!}, \\
A_{n,k} &= \binom{n+k-2}{k-1} \frac{(2n-k-1)!}{(n-k)!} \frac{(n-1)!}{(2n-2)!} A_{n-1}, \\
AV_{2n+1} &= \prod_{j=0}^{n-1} (3j+2) \frac{(6j+3)!(2j+1)!}{(4j+3)!(4j+2)!}, \\
C_{2n} &= \prod_{j=0}^{n-1} (3j+1) \frac{(6j)!(2j)!}{(4j)!(4j+1)!}, \\
AVH_{2n+1} &= AV_{2\lfloor \frac{n}{2} \rfloor + 1} C_{2\lfloor \frac{n+1}{2} \rfloor}, \\
AHT_{2n} &= \prod_{j=0}^{n-1} \frac{(3j)!(3j+2)!}{((n+j)!)^2}, \\
AHT_{2n+1} &= \frac{n!}{(3n+2)!} \prod_{j=0}^{n} \frac{(3j)!(3j+2)!}{((n+j)!)^2},
\end{aligned}
\tag{94}
$$

were $A_n$ is the number of $n \times n$ alternating sign matrices (ASMs), $A_{n,k}$ is the number of $n \times n$ alternating sign matrices constrained to have a 1 at top of column $k$, $AV_{2n+1}$ is the number of vertically symmetric ASMs of size $2n+1$, $AVH_{2n+1}$ is the number of vertically and horizontally symmetric ASMs of size $2n+1$, $AHT_n$ is the number of half-turn symmetric ASMs of size $n$, and $C_{2n}$ is the number of cyclically symmetric transpose complement plane partitions of size $2n$.

The boundary loop generating functions at special values of the boundary loop weight $x$ evaluate to some combinatorial numbers, which we list here for convenience and which will assist with asymptotic calculations.

Some of the results listed here are merely observations with proofs outstanding. We have collected small-size examples in B.

**Remark 1.** We note that all values of $x$ treated in the Appendices (namely $x = 2, 1, \frac{1}{2}, 0, -1$, and the limit $x \to \pm\infty$) correspond to $r \in (0, 3)$ taking half-integer values (namely $r = \frac{1}{2}, 1, \frac{3}{2}, 2, \frac{5}{2}$, and the limits $r \to 0^+$ and $r \to 3^-$). It is conceivable that the special role of $r \in \mathbb{N}$ may be accounted for by the representation theory of the one-boundary Temperley-Lieb algebra, according to which the boundary conditions corresponding to the BCC operator $\phi_{r,r}$ are expressible within the usual TL algebra in terms of a Jones-Wenzl projector that symmetrises the first physical strand with $r-1$ extra ghost strands [11]. (See also [44] for an equivalent description in terms of boundary integrability, still for $r \in \mathbb{N}$.) We also note that half-integer Kac labels of BCC operators are ubiquitous in the CFT of loop models [34], and have appeared recently in the boundary integrability framework as well [45].

## A.1  Special values of $F_{2n}^{(\mathrm{per})}(x)$

At $x = \pm 1$ the evaluation of $F_{2n}^{(\mathrm{per})}$ is given by

$$
F_{2n}(1) = 1, \qquad F_{2n}(-1) = \begin{cases} 0, & n \text{ even}, \\ \dfrac{-AV_n^2}{A_n}, & n \text{ odd}. \end{cases}
\tag{95}
$$

The case $F_{2n}(-1)$, $n$ odd, was conjectured in [46] and proved in [47]. In the current setting it follows directly from Kummer's theorem [48] for the hypergeometric series (45). The top and bottom coefficients are known to be

$$\lim_{x\to 0} x^{-1} F_{2n}(x) = \frac{A_{n-1}}{A_n} = \lim_{x\to\infty} x^{-n} F_{2n}(x). \tag{96}$$

We also have that

$$F_{2n}(2) = \frac{(2n)!}{2n!} \frac{3\frac{n}{2}!}{\frac{3n}{2}!}, \qquad F_{2n}(\tfrac{1}{2}) = 2^{-n-1} \frac{(2n)!}{2n!} \frac{3\frac{n}{2}!}{\frac{3n}{2}!}. \tag{97}$$

The second is a result from Bailey [48] for the hypergeometric series (45), and follows from Kummer's theorem after an Euler transformation. Note that the first result is related to the second by the property $F(x) = x^{n+1} F(1/x)$. If $n$ is odd, they can be expressed as

$$F_{2n}(2) = \frac{2^{2n-1} AV_n^2}{A_n}, \qquad F_{2n}(\tfrac{1}{2}) = \frac{2^{n-2} AV_n^2}{A_n}. \tag{98}$$

## A.2 Special values of $F_{2n+1}^{(\mathrm{per})}(x)$

Until now we have avoided mention of the odd-sized periodic case, because of a lack of results. However we collect here some observations at special values of $x$, and hope this will assist in finding the equivalent expression to (31) for this case.

The sum rule $Z_{2n+1} = \sum_\alpha \psi_\alpha$ of the odd-sized periodic ground state is [49, Section 2]

$$Z_{2n+1} = \mathrm{AHT}_{2n+1}, \tag{99}$$

and as usual, $F_{2n+1}(1) = 1$.

The coefficient in $F_{2n+1}(x)$ of the highest power of $x$ is $\psi_{\alpha_0}/Z_{2n+1}$, since $\alpha_0$ is the only link pattern that can give $n$ loops when paired with $\alpha_0$. Thus [23, Conjecture 9] gives the following.

**Conjecture 2.**

$$\lim_{x\to\infty} \frac{F_{2n+1}(x)}{x^n} = \frac{A_n^2}{\mathrm{AHT}_{2n+1}}. \tag{100}$$

$F_{2n+1}(0)$ gives the constant term, which is the normalised sum of all components whose corresponding link patterns produce no loops when paired with $\alpha_0$.

**Conjecture 3.**

$$F_{2n+1}(0) = \frac{\mathrm{AHT}_{2n}}{\mathrm{AHT}_{2n+1}}. \tag{101}$$

If there is a way to show that the sum of these components is equal to the sum rule for the punctured even periodic model, then this conjecture will be equivalent to one in [50] (in that article the punctured model is referred to as distinct connectivities, or "DC").

Finally we have

**Conjecture 4.**

$$F_{2n+1}(-1) = \frac{\mathrm{AV}_{2n+1}}{\mathrm{AHT}_{2n+1}}, \qquad F_{2n+1}(2) = \frac{2^{2n} \mathrm{AV}_{2n+1}}{\mathrm{AHT}_{2n+1}}. \tag{102}$$

We currently have no explanation for these observations.

## A.3 Special values of $F_{2n}^{(\mathrm{refl})}(x)$

**Proposition 4.**

$$F_{2n}(1) = 1, \tag{103}$$

$$\lim_{x \to 0} \frac{F_{2n}(x)}{x} = \frac{1}{\mathrm{AV}_{2n+1}} \frac{2^{2-n}3(2n-2)!(2n-1)!}{(n-1)!^3 n!^2 (3n)!} \prod_{i=1}^{n-1} \frac{(3i+1)!(3i+3)!(4n+2i-2)!}{(2i-1)!(3n+3i)!(2n+i-1)!}, \tag{104}$$

$$\lim_{x \to \infty} \frac{F_{2n}(x)}{x^n} = \frac{\mathrm{C}_{2n}}{\mathrm{AV}_{2n+1}}. \tag{105}$$

The product formula for $x \to 0$ is a result of applying [51, eq (2.19)]. These numbers are also found in [52] (see (3.20) with $N = 1$, $L$ odd), and conjecturally given a different product formula there. This formula is conjectured to be the sum of all components in an odd-sized system (size $2n-1$ in our notation) for which the unpaired link of the link pattern is at site 1. The coefficient of $x$ in $F_L(x)$ is simply the component $\psi_\alpha$ where $\alpha$ has pairings $(1, L)$ and $(2i, 2i+1)$, $i = 1, \ldots, n-1$. This leads one to suspect that there must be a relationship between these.

The value for $x \to \infty$ comes from the Lindstrom–Gessel–Viennot-type determinant for $C_n$, see [53, 54].

**Conjecture 5.**

$$F_{2n}(-1) = \frac{(-1)^n \mathrm{AVH}_{2n+1}^2}{\mathrm{AV}_{2n+1}}, \qquad F_{2n}(2) = \frac{\mathrm{AHT}_{2n}}{\mathrm{AV}_{2n+1}}, \qquad F_{2n}(\tfrac{1}{2}) = \frac{2^{-n}\mathrm{A}_n^2}{\mathrm{AV}_{2n+1}}. \tag{106}$$

## A.4 Special values of $F_{2n+1}^{(\mathrm{refl})}(x)$

**Proposition 5.**

$$F_{2n+1}(1) = 1, \qquad \lim_{x \to \infty} \frac{F_{2n+1}(x)}{x^n} = \frac{\mathrm{AV}_{2n+1}}{\mathrm{C}_{2n+2}}, \qquad F_{2n+1}(0) = \frac{\mathrm{AV}_{2n+1}}{\mathrm{C}_{2n+2}}. \tag{107}$$

The cases $x \to \infty$ and $x = 0$ again come from the Lindstrom–Gessel–Viennot-type determinant for $AV_n$, see [32, 53].

**Conjecture 6.**

$$F_{2n+1}(-1) = \begin{cases} \dfrac{\mathrm{AV}_{n+1}^4}{\mathrm{C}_{2n+2}}, & n \text{ even}, \\ 0, & n \text{ odd}, \end{cases} \qquad F_{2n+1}(2) = \frac{\mathrm{AHT}_{2n+1}}{\mathrm{C}_{2n+2}}, \qquad F_{2n+1}(\tfrac{1}{2}) = \frac{2^{-n}\mathrm{AHT}_{2n+1}}{\mathrm{C}_{2n+2}}. \tag{108}$$

The observation for $x = -1$ was also made in [32] (see eq. (4.5) and the discussion around eq. (4.7) of that paper). The observations for $x = 2$ and $x = \frac{1}{2}$ are related by the property $F(x) = x^n F(\frac{1}{x})$.

# B Results for small sizes

We give here small size ($L \le 14$) examples of $F_L(x)$ multiplied by the normalisation $Z_L$ for all cases (including odd periodic).

### B.1 Periodic, $L = 2n$

| $L$ | $Z_L F_L(x)$ | $Z_L$ | $Z_L F_L(-1)$ | $Z_L F_L(2)$ |
|---|---|---|---|---|
| 2 | $x$ | 1 | $-1$ | 2 |
| 4 | $x + x^2$ | 2 | 0 | 6 |
| 6 | $2x + 3x^2 + 2x^3$ | 7 | $-1$ | 32 |
| 8 | $7x + 14x^2 + 14x^3 + 7x^4$ | 42 | 0 | 294 |
| 10 | $42x + 105x^2 + 135x^3 + 105x^4 + 42x^5$ | 429 | $-9$ | 4608 |
| 12 | $429x^6 + 1287x^5 + 2002x^4 + 2002x^3 + 1287x^2 + 429x$ | 7436 | 0 | 122694 |
| 14 | $7436x + 26026x^2 + 47320x^3 + 56784x^4 + 47320x^5 + 26026x^6 + 7436x^7$ | 218348 | $-676$ | 5537792 |

### B.2 Periodic, $L = 2n + 1$

| $L$ | $Z_L F_L(x)$ | $Z_L$ | $Z_L F_L(-1)$ | $Z_L F_L(2)$ |
|---|---|---|---|---|
| 3 | $2 + x$ | 3 | 1 | 4 |
| 5 | $10 + 11x + 4x^2$ | 25 | 3 | 48 |
| 7 | $140 + 232x + 167x^2 + 49x^3$ | 588 | 26 | 1664 |
| 9 | $5544 + 12182x + 12617x^2 + 7097x^3 + 1764x^4$ | 39204 | 646 | 165376 |
| 11 | $622908 + 1699522x + 2262448x^2 + 1804988x^3 + 849080x^4 + 184041x^5$ | 7422987 | 45885 | 46986240 |
| 13 | $198846076 + 646978332x + 1044949413x^2 + 1059015059x^3 + 703061958x^4 + 286853502x^5 + 55294096x^6$ | 3994998436 | 9304650 | 38111846400 |

### B.3 Reflecting, $L = 2n$

| $L$ | $Z_L F_L(x)$ | $Z_L$ | $Z_L F_L(-1)$ | $Z_L F_L(2)$ |
|---|---|---|---|---|
| 2 | $x$ | 1 | $-1$ | 2 |
| 4 | $x + 2x^2$ | 3 | 1 | 10 |
| 6 | $4x + 11x^2 + 11x^3$ | 26 | $-4$ | 140 |
| 8 | $50x + 171x^2 + 255x^3 + 170x^4$ | 646 | 36 | 5544 |
| 10 | $1862x + 7540x^2 + 14196x^3 + 14858x^4 + 7429x^5$ | 45885 | $-1089$ | 622908 |
| 12 | $202860x + 944119x^2 + 2107417x^3 + 2828644x^4 + 2301150x^5 + 920460x^6$ | 9304650 | 81796 | 198846076 |
| 14 | $64080720x + 335905878x^2 + 859371991x^3 + 1374229792x^4 + 1453822999x^5 + 971405460x^6 + 323801820x^7$ | 5382618660 | $-19536400$ | 180473355920 |

### B.4 Reflecting, $L = 2n + 1$

| $L$ | $Z_L F_L(x)$ | $Z_L$ | $Z_L F_L(-1)$ | $Z_L F_L(2)$ |
|---|---|---|---|---|
| 3 | $1 + x$ | 2 | 0 | 3 |
| 5 | $3 + 5x + 3x^2$ | 11 | 1 | 25 |
| 7 | $26 + 59x + 59x^2 + 26x^3$ | 170 | 0 | 588 |
| 9 | $646 + 1837x + 2463x^2 + 1837x^3 + 646x^4$ | 7429 | 81 | 39204 |
| 11 | $45885 + 156107x + 258238x^2 + 258238x^3 + 156107x^4 + 45885x^5$ | 920460 | 0 | 7422987 |
| 13 | $9304650 + 36756435x + 71760049x^2 + 88159552x^3 + 71760049x^4 + 36756435x^5 + 9304650x^6$ | 323801820 | 456976 | 3994998436 |

## C  Asymptotics of $F_L^{(\text{refl})}(x)$ for $x = -1, 0, \frac{1}{2}$, and 2

Conjecture 1 is supported by the following results for special values of $x$ according to A.3 and A.4, whose asymptotics can be derived from that of the Barnes $G$-function, see (48). In the following we denote by $A$ the Glaisher constant.

First recall (24),

$$\tilde{F}_L^{(\text{refl})}(x) = \exp\left(ng_0(x) + \log(n)g_1(x) + g_2(x) + n^{-1}g_3(x) + \dots\right). \tag{109}$$

For $x = 1$ ($r = 1$) we have that $\tilde{F}_L(1) = 1$ and hence $g_j(1) = 0 \ \forall j$. In addition we have results at $x = -1, 0, \frac{1}{2}$ and 2 ($r = \frac{5}{2}, 2, \frac{3}{2}$ and $\frac{1}{2}$).

### C.1  $L$ even

Here we list the asymptotics for $L = 2n$ obtained from the results in A.3. From Conjecture 5 and Proposition 4 we find

$$g_0(-1) = \log\left(\frac{2}{3\sqrt{3}}\right), \quad g_1(-1) = \frac{1}{8}, \quad g_2(-1) = \frac{1}{24} + \log\left(\frac{3^{\frac{11}{24}}\Gamma(\frac{1}{3})}{2^{\frac{1}{18}}(\pi A)^{\frac{1}{2}}}\right), \tag{110}$$

$$g_0(0) = \log\left(\frac{16}{27}\right), \quad g_1(0) = -\frac{1}{2}, \quad g_2(0) = \log\left(\frac{3}{(2\pi)^{\frac{1}{2}}}\right), \tag{111}$$

$$g_0(2) = \log\left(\frac{8}{3\sqrt{3}}\right), \quad g_1(2) = \frac{1}{8}, \quad g_2(2) = -\frac{3}{8} + \log\left(\frac{\Gamma(\frac{1}{3})}{3^{\frac{1}{24}}2^{\frac{1}{18}}(\pi A)^{\frac{1}{2}}}\right), \tag{112}$$

$$g_0(\tfrac{1}{2}) = \log\left(\frac{4}{3\sqrt{3}}\right), \quad g_1(\tfrac{1}{2}) = -\frac{5}{24}, \quad g_2(\tfrac{1}{2}) = \frac{1}{24} + \log\left(\frac{2^{\frac{7}{9}}\pi^{\frac{1}{4}}}{3^{\frac{7}{24}}(A\Gamma(\frac{1}{6}))^{\frac{1}{2}}}\right). \tag{113}$$

### C.2 *L* **odd**

Here we list the asymptotics for $L = 2n+1$ obtained from the results in A.3. From Conjecture 6 and Proposition 5 we find

$$g_0(-1) = \log\Big(\frac{2}{3\sqrt{3}}\Big), \qquad g_1(-1) = -\frac{3}{8}, \qquad g_2(-1) = \frac{1}{24} + \log\Big(\frac{2^{\frac{25}{9}}\pi}{3^{\frac{25}{24}}\Gamma(\frac{1}{3})^2 A^{\frac{1}{2}}}\Big), \qquad (114)$$

$$g_0(0) = \log\Big(\frac{16}{27}\Big), \qquad g_1(0) = -\frac{1}{6}, \qquad g_2(0) = \log\Big(\frac{2^{\frac{17}{6}}\pi^{\frac{1}{2}}}{3^{\frac{3}{2}}\Gamma(\frac{1}{3})}\Big), \qquad (115)$$

$$g_0(2) = \log\Big(\frac{8}{3\sqrt{3}}\Big), \qquad g_1(2) = -\frac{1}{24}, \qquad g_2(2) = \frac{1}{24} + \log\Big(\frac{2^{\frac{16}{9}}}{3^{\frac{25}{24}}A^{\frac{1}{2}}}\Big), \qquad (116)$$

$$g_0(\tfrac{1}{2}) = \log\Big(\frac{4}{3\sqrt{3}}\Big), \qquad g_1(\tfrac{1}{2}) = -\frac{1}{24}, \qquad g_2(\tfrac{1}{2}) = \frac{1}{24} + \log\Big(\frac{2^{\frac{16}{9}}}{3^{\frac{25}{24}}A^{\frac{1}{2}}}\Big). \qquad (117)$$

## D Periodic asymptotics, lower order terms

The computations of $f_j(x)$, $j > 2$, are completely analogous to the case $j = 2$. One first gets an expression for $f_j'(x)$ in terms of $f_k'(x)$ with $k = 0, 1, \ldots, j-1$ and $f_{j-1}''(x)$, all of which are known. Integrating one obtains $f_j(x)$, and the constant of integration is chosen to match the initial conditions in (47). Rewriting in terms of $r$, one obtains $-S_{-j+1}(r)$.

The results can be written as follows for $x \geq -1$ (for $x = -1$ the results are only valid for odd $n$, as in Proposition 1),

$$S_{-1} = -\frac{5}{36}\cos^2\Big(\frac{\pi r}{2}\Big),$$

$$\frac{S_{-2}}{S_{-1}} = \frac{1}{2}\cos(\pi r),$$

$$\frac{S_{-3}}{S_{-1}} = \frac{1}{864}\big(-15 - 10\cos(\pi r) + 221\cos(2\pi r)\big),$$

$$\frac{S_{-4}}{S_{-1}} = \frac{1}{576}\big(-5 - 51\cos(\pi r) - 5\cos(2\pi r) + 113\cos(3\pi r)\big),$$

$$\frac{S_{-5}}{S_{-1}} = \frac{1}{248832}\big(225 - 1826\cos(\pi r) - 37952\cos(2\pi r) - 1758\cos(3\pi r) + 49695\cos(4\pi r)\big),$$

$$\frac{S_{-6}}{S_{-1}} = \frac{1}{497664}\big(1605 + 22102\cos(\pi r) - 1760\cos(2\pi r) - 135990\cos(3\pi r) - 3365\cos(4\pi r) + 125920\cos(5\pi r)\big). \qquad (118)$$

The results for $x < -1$ are obtained from the above by replacing $r$ with $r - 3$.

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
