# Peer review of "Finite-size corrections for universal boundary entropy in bond percolation"

_SciPost Physics, doi:SciPost Phys. 1, 012 (2016)_

## Round 1 · Referee Report · Anonymous · 2016-10-31

Strengths

1) New and interesting results
2) Stimulating new researches in the field
3) Cross-fertilization beween integrability and combinatorics
4) Well written paper

Weaknesses

none

Report

The paper investigates the boundary entropy for bond percolation on the square lattice (that is the Temperley-Lieb loop model at the `combinatorial' point $\beta=1$), obtaining explicit exact results on a semi-infinite strip or cylinder of width L.

The asymptotic behaviour of the obtained expressions at large L is analysed.

In the case of the even L cylinder, results are obtained in full rigour, and agrees with Conformal Field Theory predictions. In the case of the strip, obtained expressions are conjectural, but strongly supported by numerics, and in full agreement with predictions based on Conformal Field Theory, concerning the emergence of universal logarithmic corrections, induced by the presence of corners in the strip.

The paper rises several interesting questions, stimulating further investigations, in particular:
- working out the combinatorics of the odd-L model on the cylinder;
- understanding, at combinatorial level, the observed equivalence between two completely different refined countings of the configurations of the model on the even-L cylinder;
- providing a rigorous derivation of the large L behaviour of the boundary entropy of the model on the strip.

The results presented, and the questions rised, are of great interest in the field of exactly solvable models, as well as in combinatorics.

Requested changes

I would like to point the attention of authors to the fact that the leading term in the asymptotics of $\tilde{F}^{(per)}$, denoted by $f_0(x)$ in Proposition 1, page 7, was
first evaluated in
http://dx.doi.org/10.1137/080730639 , or
http://xxx.arxiv.org/pdf/0803.2697 , see Section 4 therein.

This should be mentioned.Also, I noticed a couple of typos:
1) In eq. (10), $\psi_L$ should be capitalized (twice)
2) Last sentence of page 7, verb is missing

  • validity: high
  • significance: high
  • originality: high
  • clarity: high
  • formatting: excellent
  • grammar: excellent

Author:  Anita Ponsaing  on 2016-11-08  [id 71]

(in reply to Report 1 on 2016-10-31)

Thank you for your kind comments, and for your suggestions, in particular alerting us to the omission of the reference. We will make the suggested changes in our revision.

---

## Round 1 · Referee Report · Anonymous · 2016-11-12

Strengths

1- The paper presents interesting and original new results about the boundary entropy for the model of bond percolation.
2- The authors place their results in the context of the current knowledge in this field and correctly cite the relevant literature.
3- The distinction between rigorously proved results and conjectures is clear.
4- The main results, both conjectures and rigorously proved results, are supported by high-precision numerical data.
5- The presentation of the ideas is coherent and clear.
6- The text is well written with only few typos.

Weaknesses

1- In carrying out the proofs of their expressions for the boundary entropy, the authors use many intermediate results coming from previous articles, so the paper is not self-contained and it can be hard for the reader to have a complete picture of the proofs of the main results.

Report

In this paper, the authors compute the boundary entropy for the loop model corresponding to bond percolation. They obtain finite-size results for the corresponding scalar product, and have done so mostly by gathering results scattered across the literature. They also provide an asymptotic analysis of their results, using an exact derivation for the periodic boundary conditions, and arguments of conformal field theory for the strip boundary conditions. The results are in agreement with the general prediction of conformal field theory as well as with numerical data.

My review of this paper is very positive. There are relatively few examples where such expressions can be obtained for finite-size systems and I therefore believe that the paper makes an important contribution.

The paper is well written in general and is easy to read. After studying the manuscript carefully, I have come up with a list of questions, suggestions and minor points to be fixed.

Requested changes

1- There seems to be an inconsistency in eq. (10). If < | > is indeed the usual Temperley-Lieb bilinear form with the loop weight set to 1, then <\alpha | \alpha’ > = 1 for any \alpha, \alpha’. This implies that <\psi_L | \psi_L> = (\sum_{\alpha} \phi_\alpha) * (sum_{\alpha} 1). It is therefore the sum of the components of |\psi_L> times the dimension of the vector space. One way to fix this is to define Z_L to be <\alpha_0 | \psi_L> instead, in which case it indeed equals the sum of the components.

2- In eq. (12), the first equality should read F_L(x) = \frac{1}{Z_L}\sum_\alpha \phi_\alpha <\alpha_0| \alpha>_x. (Two things are missing: the normalization Z_L and the \phi_\alpha inside the sum.)

3- It might be useful to give this definition of F_L at the start of eq. (12): F_L = <\alpha_0 | \psi_L >_x / <\alpha_0 | \psi_L >_{x=1}. This will help the reader understand later why F_L = 1 for x = 1.

4- As correctly discussed in Section 4.1, the asymptotic expansion of F^{per}_{2n}(x) sometimes requires the presence of a minus sign. This should be incorporated in eq. (20) in some way, otherwise the proposition is not quite correct.

5- Below eq. (41), the text states that the determinant in eq. (40) can be written as x^{n-1} S(2n,n-1,1/x). After double-checking, I find that this determinant is in fact equal to x^{n} S(2n,n-1,1/x). This is consistent with the expression for F^{(refl)}_2n(x) given below eq. (42).

6- Below eq. (41), the authors refer to some result of [29] specified to \beta = 1, but the corresponding result in reference [29] is in fact written in terms of a parameter \tau which should be specified to 1. The same comment applies to the sentence three lines below (44).

7- Below eq. (43), it would be useful to state that the values of j of the a_j in the integer sequences in Q_n are in the range 1, ..., n.

8- Three lines below eq. (44): in “the determinant in (41) ... appears in [29, eq. (6.19)] as S(2n+1,n|1/x)”, I believe a factor of x^n should be added in front of S(2n+1,n|1/x).

9- In (45), I do not understand why the prefactor is written in such a complicated fashion, in terms of Gamma functions with half-integer arguments. Moreover, if I am not mistaken, the expression given by the authors for this prefactor is incorrect. According to my calculation, the prefactor should be (2n-1)! (2n-2)!/(n-1)!/(3n-2)!, which is equivalently written as 4^{n-1} Gamma[n+1/2] Gamma[2n-2]/Gamma[1/2]/Gamma[3n-2].

10- Is there any reason to expect, from the lattice model, the existence of a differential equation (eq. (46)) satisfied by the function F^{(per)}_{2n}? Do you suspect that a similar differential equation could exist for the reflecting case?

11- In eq. (47), F(-1) takes different values depending on the parity of n. These two expressions clearly have different asymptotics. While sections 4.1.1 and 4.1.2 solve for the asymptotics for x > -1 and x < -1, eqs. (21)-(23) are claimed to hold also at x = -1. Are the asymptotics at x = -1 given by eqs. (21)-(23) only valid for n odd? Please clarify this in the text.

12- According to eq. (108), the next term in the expansion on the first line of eq. (48) is n^{-2}*5/72. The O(n^{-3}) in the first line of eq. (48) is therefore incorrect.

13- In eq. (52), the logarithm poses a problem for x -> - \infty. Not only because the argument of the logarithm is negative. Eq. (52) is not quite compatible with eq. (63) because a sign is missing. I think the solution is to add an absolute value, so that eq. (52) reads: f_0(0) = \lim_{x \rightarrow \pm \infty} ( f_0(x) - log |x| ) = \log (16/27).

14- The expressions (21)-(23) for exp(f_0), exp(f_1) and f_2 in the two regimes have simple behaviors under the transformation r -> 3-r which corresponds to x -> 1-x. Do you have an explanation for this unexpected feature from the lattice model?

15- In the conclusion, the authors state that the results for the periodic boundary conditions precisely agree with the predictions of conformal field theory. I believe a more thorough analysis is warranted here. In particular, the (possible) universality of the functions f_j should be discussed. For instance, can f_1 be written in terms of the central charge and the conformal weights? Or does it reproduce an expression known in the literature? In other words, is an analysis similar to that of Section 4.2 possible for the periodic boundary conditions as well?

16- Can you comment on the possible extension of the results for periodic boundary conditions to the representation of the periodic Temperley-Lieb with distinct connectivities (wherein the top of the cylinder is “punctured”)?

17- Are the results of eq. (108) valid for the regime 0<r<5/2 only? This should be clarified. Do you also have the results in the second regime? Can these be obtained from the results in the first regime by replacing r by r-3?

---

## Round 2 · Author Response

Dear Editor,

We enclose our revised manuscript and give detailed responses to the referee reports below.

Reply to referee 1:

Thank you for your kind comments. We have included the proposed reference and fixed up the typos.

Reply to referee 2:

Thank you for your detailed and valuable report. As several results are scattered in existing literature, we made a conscious choice to try and unify these by explicitly indicating links to other papers and explain differences in notation.

1 To avoid a precise explanation which would interrupt the flow we now omit the bra-ket notation in (10) and simply define the norm as the sum over components.

2 Fixed

3 Added, thanks for this suggestion

4 We have included a sign factor $\varepsilon_{n,x}$ in (20) and (24)

5 Fixed

6 Fixed in both places

7 Fixed -- come to think of it $a_1$ should always equal 1, so $j$ is in ${2,...,n}$.

8 Fixed

9 Thank you for the simpler expression, which has been included... however we believe that our original expression was correct and equal to this one.

10 Interesting question which had crossed our minds as well. We suspect that there should be a differential equation for the reflecting case, but preliminary attempts to find one indicate that it will not be as simple as the hypergeometric equation for the periodic case. We leave this for future study.

11 Yes. Thanks for catching this. A sentence has been added to Proposition 1.

12 Fixed

13 Fixed. Thanks for catching this!

14 This is an astute observation and a good question to which we currently don't have a clear answer.

15 We have added a whole new section (4.3) discussing the CFT argument for the even periodic case.

16 The case of distinct connectivities is indeed very interesting and we believe an extension to this case is possible. Like the odd periodic case we refer this to future study.

17 Yes, and yes. Fixed with an extra sentence below what is now equation (118).

---

## Round 2 · List of Changes

- Sec 1, para 5: Added sentence about the entanglement entropy of the stochastic raise-and-peel model
- Eq (10): Removed bra-ket notation (Ref 2 point 1)
- Eq (10): Capitalised $\Psi$, cf Ref 1 point 1
- Eq (12): Changed, cf Ref 2 points 2&3
- Sec 1.3, para after eq (12): Added "precisely"
- Eq (20): Added sign factor, cf Ref 2 point 4
- Prop 1: Added caveat, cf Ref 2 point 11
- Sec 2.2.1, final para: Added reference, cf Ref 1 requested change
- Eq (24): Added sign factor, cf Ref 2 point 4
- Sec 2.2.2, final para: Added `be', cf Ref 1 point 2
- Proof of Prop 3, para 1: changed $x^{n-1}$ to $x^n$, cf Ref 2 point 5
- Proof of Prop 3, para 1: added $\tau$, cf Ref 2 point 6
- Proof of Prop 3, para 2: changed specification of $a_j$, cf Ref 2 point 7
- Proof of Prop 3, final para: added $x^n$, cf Ref 2 point 8
- Proof of Prop 3, final para: added $\tau$, cf Ref 2 point 6
- Eq (45): changed prefactor, cf Ref 2 point 9
- Sec 4.1, before eq (46): Added reference
- Eq (48): changed $\mathcal{O}(n^{-3})$ to $\mathcal{O}(n^{-2})$, cf Ref 2 point 12
- Eq (52): changed $\log(x)$ to $\log|x|$, cf Ref 2 point 13
- Section 4.3 has been added, explaining conformal reasoning behind the results for the even periodic case. Cf Ref 2 point 15
- Sec 5, para 2: Added some words about precision of numerical data
- App D: Incorporated information for $x\leq-1$, cf Ref 2 point 17

You are currently on this page

Resubmission 1610.04006v2 on 24 November 2016

---

## Editorial Decision

published